# The Newfoundland and Labrador mosaic founder population descends from an Irish and British diaspora from 300 years ago

Edmund Gilbert [1,2✉], Heather Zurel [3], Margaret E. MacMillan[3], Sedat Demiriz [3], Sadra Mirhendi [3], Michael Merrigan[4], Seamus O'Reilly[4], Anne M. Molloy[5], Lawrence C. Brody[6], Walter Bodmer [7], Richard A. Leach[3], Roderick E. M. Scott[3], Gerald Mugford[3], Ranjit Randhawa[3], J. Claiborne Stephens[3], Alison L. Symington [3], Gianpiero L. Cavalleri[1,2] & Michael S. Phillips [3]

The founder population of Newfoundland and Labrador (NL) is a unique genetic resource, in part due to its geographic and cultural isolation, where historical records describe a migration of European settlers, primarily from Ireland and England, to NL in the 18th and 19th centuries. Whilst its historical isolation, and increased prevalence of certain monogenic disorders are well appreciated, details of the fine-scale genetic structure and ancestry of the population are lacking. Understanding the genetic origins and background of functional, disease causing, genetic variants would aid genetic mapping efforts in the Province. Here, we leverage dense genome-wide SNP data on 1,807 NL individuals to reveal fine-scale genetic structure in NL that is clustered around coastal communities and correlated with Christian denomination. We show that the majority of NL European ancestry can be traced back to the south-east and south-west of Ireland and England, respectively. We date a substantial population size bottleneck approximately 10-15 generations ago in NL, associated with increased haplotype sharing and autozygosity. Our results reveal insights into the population history of NL and demonstrate evidence of a population conducive to further genetic studies and biomarker discovery.

[1] School of Pharmacy and Biomolecular Sciences, Royal College of Surgeons in Ireland, Dublin, Ireland. [2] FutureNeuro SFI Research Centre, Royal College of Surgeons in Ireland, Dublin, Ireland. [3] Sequence Bioinformatics, Inc., St. John's, Newfoundland and Labrador, Canada. [4] Genealogical Society of Ireland, Dún Laoghaire, Ireland. [5] School of Medicine, Trinity College, Dublin, Ireland. [6] Genome Technology Branch, National Human Genome Research Institute, National Institutes of Health, Bethesda, MD 20892, USA. [7] Weatherall Institute of Molecular Medicine, John Radcliffe Hospital, Oxford, UK. ✉email: edmundgilbert@rcsi.ie

Newfoundland and Labrador (NL) is the most eastern Canadian province. It is comprised of Labrador on the Canadian mainland, and the island of Newfoundland located in the north Atlantic. The European-ancestry population is primarily derived from Irish and English settlers[1] who came to NL in the 18th and 19th centuries[2]. This NL population was historically small, with a census reporting 74,094 individuals in 1836[2], but has expanded to approximately 520,000 (Statistics Canada) today. Historical records suggest that the Irish, predominantly Catholic settlers, came from communities in the southwestern Irish counties of Waterford, Wexford, south Kilkenny, southeast Tipperary, and southeast Cork[3]. In the case of the English, the mainly Protestant settlers can be traced back to the counties of Dorset and Devon as well as the fishing ports such as Dartmouth, Plymouth, and Southampton in southwestern England[3,4]. Prior to and after European settlement of the region several Indigenous Peoples including the maritime Archaic peoples, Mi'kmaq, the Innu and the Inuit, and Beothuk, inhabited areas within the modern Province[5,6]. The modern population of NL includes peoples of Indigenous ancestry, primarily the Inuit, Innu, and Mi'kmaq. The Inuit and Innu are mainly located within Labrador, and the Mi'kmaq are found within Newfoundland, both with some level of admixture with people of European ancestry[4].

Migration of Catholic Irish and Protestant English settlers to NL peaked in the mid to late 18th century[4]. These migrants who settled into coastal communities (known as outports) were socially and geographically isolated from one another, rarely intermarrying and so experienced subsequent private or separate bottlenecks. This cultural and geographical isolation is mirrored in the genetic landscape of NL. Previous studies using blood antigen frequency or birth-record data found evidence of genetic isolation and elevated inbreeding coefficients[5,7,8], though there is conflicting evidence of extended linkage disequilibrium[1,9]. These population characteristics have been successfully leveraged to identify both recessive and dominant Mendelian traits[10–14]. Recently, analysis of genome-wide SNP-array genotype data from 494 NL individuals confirmed evidence of genetic isolation in the NL population, and that the broad NL population structure could be described in terms of; (i) indigenous American ancestry and, (ii) Catholic versus Protestant background. However, with a set of 494 individuals, the extent of population structure within these three backgrounds was not explored in depth, nor the source of putative British and Irish ancestry examined.

NL is therefore a genetically understudied community, and potentially of great value to genetic mapping efforts. Isolated populations with a history of bottlenecks are valuable communities for genetic mapping efforts as founder effects increase the frequencies of rare clinically relevant variants[15] as well as increasing average haplotype length and general homogeneity[16]. Such efforts have been realised in other genetically isolated populations[16,17], including island communities such as Sardinia in Europe[18], the Ryukyu Archipelago south of Japan[19], or the Shetlands in northern Scotland[15]. As isolated populations are increasingly leveraged in the study of rare or ultra-rare genetic variation[15], an appreciation of their fine-scale genetic landscape is needed to account for the increased stratification of rarer genetic variation[20,21]. Although haplotype-based methods have demonstrated fine-scale genetic structure in many populations including the ancestral source populations of Britain and Ireland[22–25], such approaches are yet to be applied within the context of NL. Furthermore, applying haplotype-based approaches to the NL context could also reveal insights into the population's demographic history and isolation[26–29].

Given this context, we aimed to explore the non-Indigenous (European) settlement of the present NL population in unprecedented detail, studying a sample of 1807 individuals with NL ancestry from the Newfoundland and Labrador Genome Project (NLGP), together with ancestry source references from Britain and Ireland. We set out to; (i) characterise the fine-scale population structure in NL using haplotype-based methods and investigate how this structure relates to ancestry, religion, and geography, (ii) quantify the proportions of British and Irish ancestry in NL and map these to their regional sources in Britain and Ireland using ancestry references[23,30,31], and finally (iii) characterise the extent that a history of bottlenecks has had on the haplotype diversity of NL compared to ancestral sources in Britain and Ireland.

## Results

**Newfoundland genetic structure.** To sample the genetic landscape of NL before modern economic migration in the latter 20th century onwards, we applied principal component analysis (PCA) to genotype data from 2446 participants from the Newfoundland and Labrador Genome Project (NLGP—Sequence Bioinformatics, Inc) and world-wide ancestry references from HGDP or KGP3 (see Supplementary Figs. 1–2). We identified and defined NLGP individuals of "NL ancestry" as individuals who occupied the same ancestry space as either European or Indigenous American ancestry. We further performed sample and marker QC (see Methods), leaving a core dataset of 1807 NL individuals (the "$NL_{1,807}$" dataset) and 685,221 common SNPs for further analysis.

To investigate the fine-scale genetic structure of NL, we performed haplotype-based clustering using *fineSTRUCTURE*[32] to cluster individuals based on their haplotype sharing, as quantified by the *ChromoPainter* co-ancestry matrix. *fineSTRUCTURE* analysis identified 22 discrete clusters which summarise *fineSTRUCTURE*'s 74 clusters from its final *maximum a posteriori* (MAP) state. This $k = 22$ level of clustering combines smaller, difficult to interpret, clusters together to summarise the predominant fine-scale structure present in NL (Fig. 1a). The 22 clusters are further grouped in a dendrogram, organising clusters that share excess haplotypes together on shared branches. Most clusters show geographical stratification (Fig. 2 and Figs. S3-8 for individual plots) as well as religious stratification (Fig. 1b, Supp Data 1). The clusters exhibit comparatively high genetic differentiation, as measured by $F_{ST}$ (average $F_{ST}$: 0.00206, min $F_{ST}$: 0. 00016, max $F_{ST}$: 0.00429; Supp Data 2), for example an order of magnitude higher than what is found between equivalent *fineSTRUCTURE*-cluster estimates from Ireland or England ($F_{ST}$: 0.0003, or 0.0003 respectively)[23]. Measuring connectivity with average sharing of IBD segments supports this differentiation with comparatively low sharing between clusters not on the same *fineSTRUCTURE* branch (Supp Fig. 8) We further measure connectivity by recording this sharing as a network and find additional support of wide sharing amongst northern Avalon, Trinity and Conception Bays with outlying regions less connected to this central region (Supplementary Fig. 9).

We compared this structure to previous samples of NL ancestry reported by Zhai et al[4]. We found evidence that both datasets have the same individual four separate times (Supplementary Fig. 10) and found that our sample of NL-ancestry captures the variation in population structure (Supplementary Fig. 11) and haplotype diversity (Supplementary Fig. 12). Unfortunately due to genotyping platform differences, the common-marker-set was low (167,968 SNPs), therefore we decided not to incorporate the Zhai et al.[4] data into the wider analysis.

The first split in the *fineSTRUCTURE* dendrogram separates individuals in the south-east of NL with grandparents

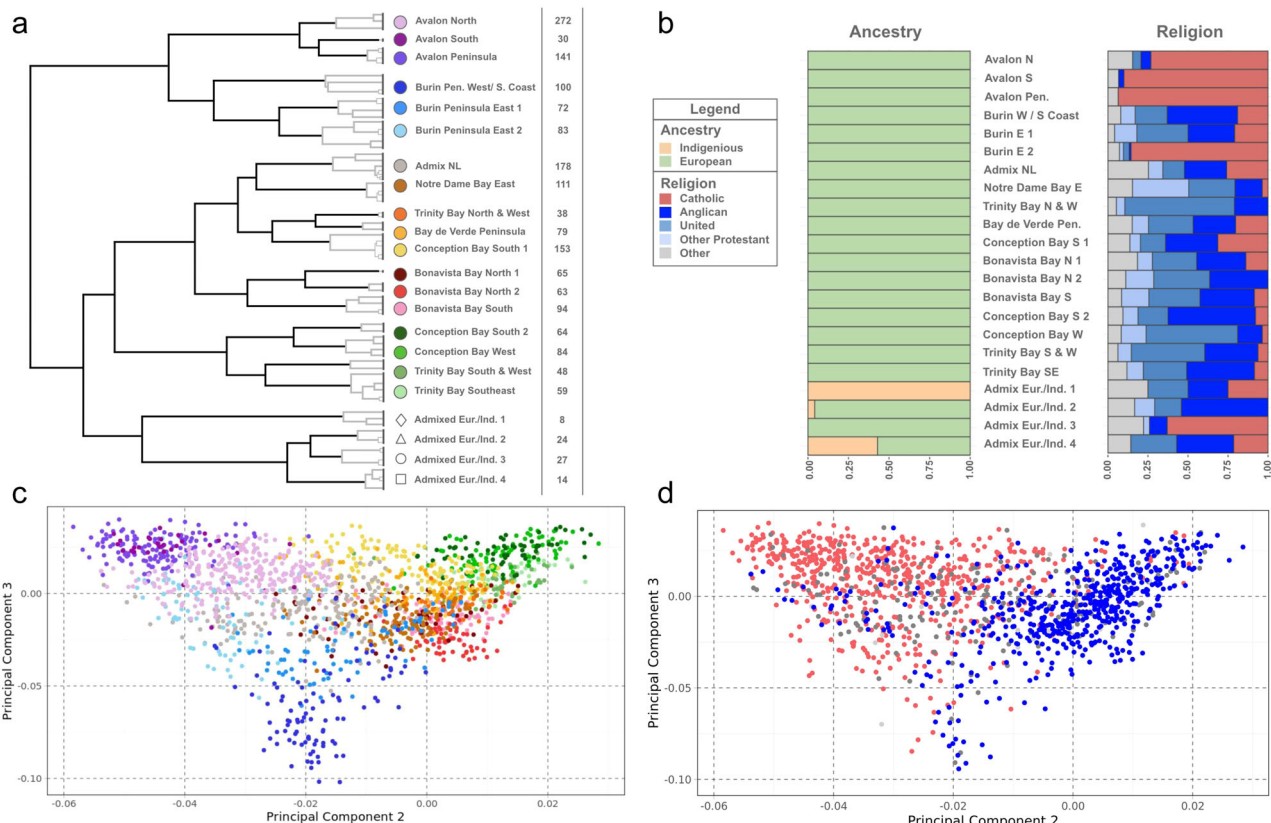

**Fig. 1 Genetic structure of Newfoundland and Labrador. a** Dendrogram of the *fineSTRUCTURE* final MAP state, showing 22 summarising clusters of NL membership. Solid branches indicate the 22 cluster branches, with grey branch-lines showing merged clusters within each of the 22 clusters. *fineSTRUCTURE* clusters are colour and shaped coded to reflect grouping on adjacent branches, with labels reflecting common geographic birthplace of members' grandparents. Cluster sizes shown to the right of cluster labels. **b** The proportions of genetic ancestry groups and religious background in each of the 22 clusters, shown in the order of A. **c** The second versus the third principal components of the *ChromoPainter* co-ancestry matrix, with individual colour and shape coded according to *fineSTRUCTURE* cluster membership. **d** The second versus the third principal components of the *ChromoPainter* co-ancestry matrix, with individual points colour coded to religious background; red indicating Christian Catholic, blue indicating Christian Protestant, and grey indicating other or unknown. All panels were plotted using the statistical computing language R[63] and the packages ggplot2 and rworldxtra.

predominantly from either Burin or the Avalon peninsula into six clusters (Fig. 2). Indeed, individuals with ancestry from south-eastern NL appear to be genetically distinct from the rest of NL in PCA (Fig. 1c) as well as clustering (Fig. 1a). When we compared the proportion of individuals in each *fineSTRUCTURE* cluster associating with various Christian denominations, we found four out of the six clusters (Fig. 1b, d) show high proportions of Catholic background. There are significant differences of religious background between genetic NL clusters as measured by $X^2$ test ($p = 0.00049$), in agreement with previous work[4]. These differences are largely driven by Catholic membership within the Avalon peninsula (the *Avalon Pen.* and *Avalon N* clusters), whose residuals contribute approximately 22% of the overall $X^2$ statistic. Furthermore, PCA of the *ChromoPainter* co-ancestry matrix supported this observation by separating individuals with a Catholic or Protestant background along PC2, echoing previous observations[4] (Fig. 1d, Supplementary Fig. 13). Evaluating median IBD-sharing versus median distance between grand-parental birthplaces for each non-Indigenous-ancestry (see below) pair of NL cluster, we see evidence that IBD shared between clusters of the same Christian denomination (Catholic or Protestant) is higher than between different Christian background (Supplementary Fig. 14) at the same geographic distance—despite low sample size of clusters. This signal is repeated specifically in the Burin peninsula (Supplementary Fig. 15) where the *Burin E 1* and *Burin E 2* clusters are interspersed.

The other major branch in the fineSTRUCTURE dendrogram separated out clusters of the remaining NL individuals and those with putative Indigenous ancestry (Fig. 1a, b). Individuals in this branch cluster with world-wide Indigenous American ancestry references in PCA (Supplementary Fig. 1), They further separate from other NL individuals along the first PC in NL-only PCA (Supplementary Fig. 16). Clusters *Admix Eur./Indig. 1* and *4* specifically present higher proportions of American ancestry components in an supervised ADMIXTURE[33] analysis using KGP3 ancestry references assuming $k = 5$ ancestry components (Supplementary Fig. 17). This ancestry component is highly correlated with the KGP3 projected PC three (Pearson r = 0.91, $p = <2 \times 10^{-16}$). The two remaining clusters do not exhibit elevated levels of this component, but are grouped with putatively Indigenous ancestry clusters, and therefore difficult to further interpret. As our focus was to characterise the genetic structure of NL arising from the non-indigenous European settlers and given the lack of appropriate North American Indigenous references in addition to the small number of participants in these clusters (4% of the cohort), we made no further effort to characterise indigenous ancestry in subsequent analyses and focussed analysis on the remaining European-descent 18 clusters.

Beyond the south-eastern branch of the dendrogram, clusters demonstrate fine-scale structure among predominantly Protestant communities. These clusters align strikingly with the geographic features of individual NL bays. For example, the north-eastern

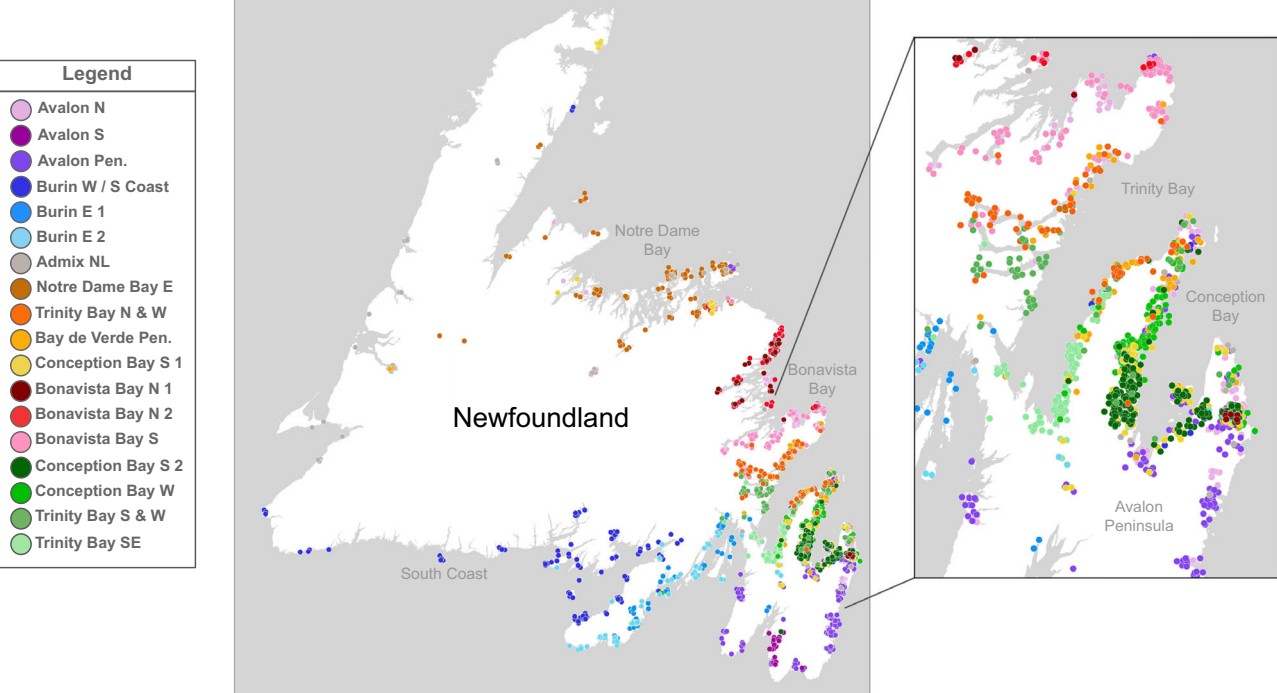

**Fig. 2 Genetic landscape of Newfoundland and Labrador.** Map of the grandparental birthplaces of individuals with colour and shape coded according to *fineSTRUCTURE* cluster. A small jitter has been introduced to aid legibility and preserve anonymity. An insert shows individual details of the Trinity and Conception Bays. Panel was plotted within photoshop, with geography boundary data sourced from Tableau.

Trinity and Conception Bays exhibit population structure not only between one another (Fig. 2) but also within the same bay (Supplementary Figs. 6 and 8). This picture is similar in the northern bays of Bonavista and Notre Dame as well. $F_{ST}$ distances between these clusters show substantial differentiation between the neighbouring communities, consistent with genetic isolation (Supplementary Data 2). This fine-scale structure arranged within individual bays is echoed in supplemental analysis where we estimate migration surfaces from discrepancies between genetic and geographic distance using the EEMS program[34] (Supplementary Note 3). Within most NL bays, this fine-scale structure is interpreted as wide areas of gene flow barriers, supportive of a landscape characterised by substantial genetic differentiation given the small geographic distances involved (see Supplemental Note 3 for further discussion). While most of these clusters are largely Protestant in religious background, there are subtle differences in denomination (Fig. 1b). The large cluster of individuals with grandparents from Trinity Bay, *Trinity Bay N&W*, show a high proportion (68%) of United Church of Canada background, as does the *Conception Bay W* cluster (57%), compared to an average of 24% elsewhere. Considering the United Church of Canada post-dates settlement, this could reflect specific communities preferentially favouring one denomination.

Elsewhere, the *Admix NL* cluster contains individuals with recent genealogies from across the island, and whose average copying vectors from the *ChromoPainter* coancestry matrix suggests haplotype sharing with different clusters across NL (Supplementary Fig. 18). Further, using the dimension-reduction methods UMAP and t-SNE[35], these individuals co-locate across the space of NL individuals (Supplementary Fig. 19-20). We infer, therefore, that the *Admix NL* cluster likely captures individuals with a mix of ancestors from across NL grouped together through *fineSTRUCTURE*'s clustering algorithm. This mixed cluster could be due to modern economic movement in the 20th century, where communities prior to the 20th century were typically

isolated. We observe a similar copying profile in the *Avalon N* cluster (Supplementary Fig. 18), which could represent the urban admixture in the metropolitan area of St John's, the province's largest city which is located to the north of the Avalon peninsula.

**Newfoundland settler ancestry.** The relative ancestry contributions from Irish and British source populations to different NL communities is largely unknown, but assumed to correlate with historical records of settlement[3] and Catholic/Protestant religious background[4,36]. To elucidate this, we used IBD-segment sharing patterns (see Methods) on a combined dataset of 1807 individuals of NL ancestry and 4,408 ancestry reference individuals from Britain and Ireland, 1,808 of whom have geographic annotation (see Supplementary Fig. 25–26 for PCA-based decomposition of this dataset). Identity-by-descent (IBD) segments are identical tracts of an ancestral haplotype shared between two individuals, shared due to common descent from a common ancestor. Due to this descent, IBD-segments are informative of both population structure[37,38] and history[27,28,39]. We first identified subcommunities within Ireland and Britain using IBD sharing network clustering (see "Methods"). We identified 26 IBD-clusters across Ireland and Britain that confirm previous *fineSTRUCTURE*-based clustering patterns (Fig. 3a). With this set of regional Irish and British reference clusters, we then leveraged an extension of a previously reported *nnls*-based approach[22] to model the proportions of IBD sharing between target and source clusters as estimated ancestry profiles.

Historical records suggest that the European settlers of NL were predominantly from south-eastern Ireland and south-western England[3]. To formally assess this mixture, we only considered IBD-segment sharing between NL individuals and Irish or British reference individuals, further considering IBD segments >3 or <15 cM in length (thereby capturing recent genealogical relationships). Our rationale being that if different NL clusters carry different proportions of Irish or British

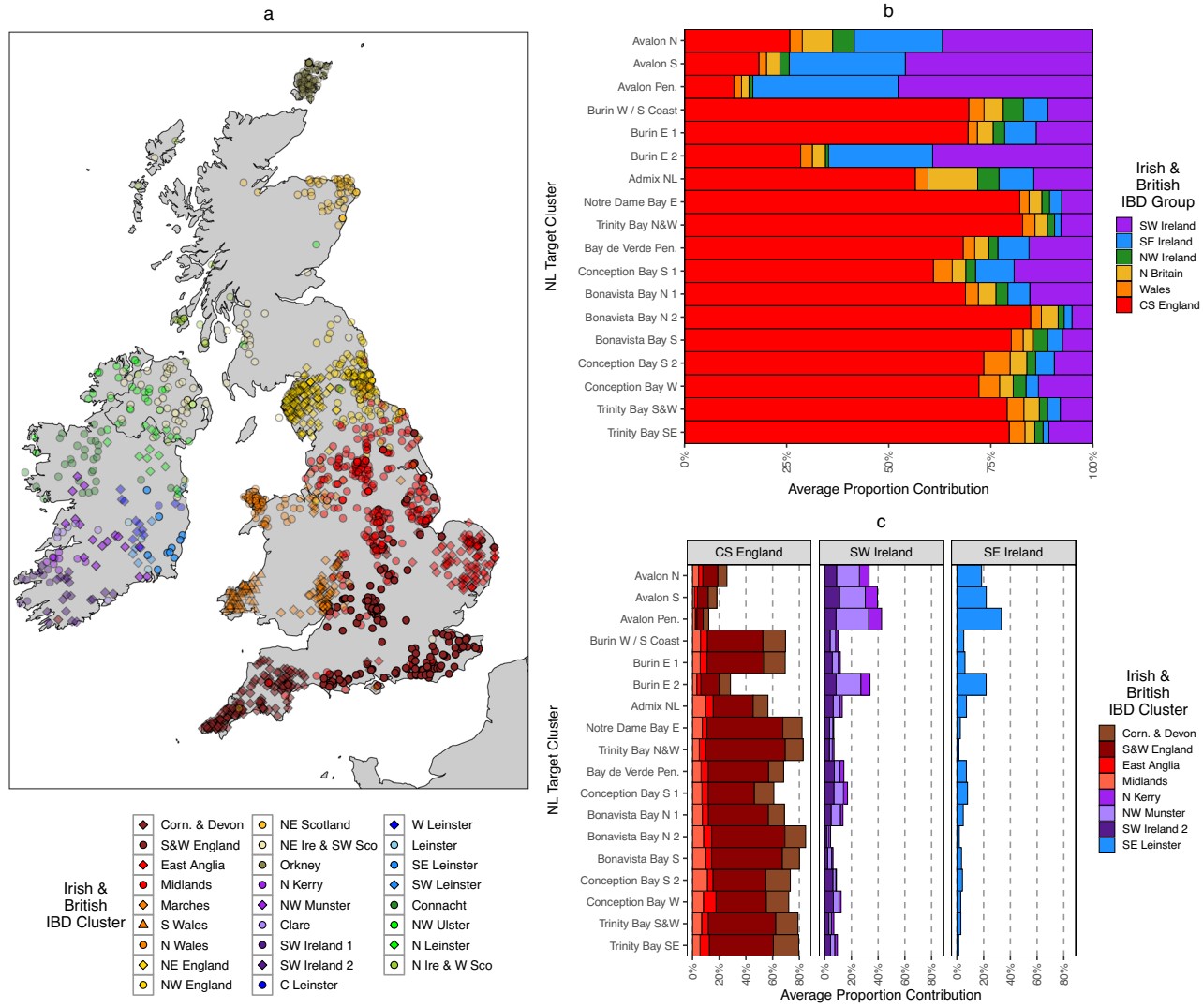

**Fig. 3 Irish and British ancestry in Newfoundland and Labrador. a** Map of Irish[23] and British[22] individuals placed according to recent mean ancestral birthplace, colour and shape coded according to IBD-network clustering. Colours are used to group hierarchically related clusters together (see "Methods"). A slight transparency for individual points has been introduced proportional to the maximal proportion that each cluster contribute to any one NL cluster, with great contributors having less transparency. **b** The summed proportions of the estimated ancestry proportions calculated from Irish or British IBD-segments shared with NL. Contributions from individual IBD clusters are summed together in groups according to Irish or British region. **c** Individual Irish or British IBD cluster contributions to NL ancestry proportions, showing only clusters that contribute >5% to any one NL cluster. All panels were plotted using the statistical computing language R[63] and the packages ggplot2 and rworldxtra.

ancestry, this will differentiate in the amount of IBD segments that they carry—thus creating different copying profiles.

We estimated the average IBD contribution from each Irish or British source IBD-cluster to each NL *fineSTRUCTURE*-cluster target, as estimated by the *nnls* method. We first summed the contributions from general Irish or British regions, such as the south-west of England, or southern Ireland (Fig. 3b, Supplementary Data 3). The results showed substantially different profiles within NL, which are largely driven by either English or southern Irish ancestry contributions. The NL clusters with a high proportion of Catholic background (typically from the Avalon peninsula) are significantly associated with increasing Irish contribution (Welch Two-Sample test $p = 0.009$), and in general a high proportion of individuals with a Catholic background are associated with a high proportion of southern Irish ancestry (Pearson $r^2 = 0.95$, $p = 7e^{-12}$) (Supplementary Figs. 27–28).

Next, we investigated if any specific IBD-cluster of Irish or British individuals were driving these English/Irish signals.

Considering individual source clusters that contribute substantially to any one NL target cluster (i.e., >5%), we indeed find specific regional affinity within the English and Irish ancestry components (Fig. 3c). The English component is driven by the *S&W England* cluster, as well as *Corn. & Devon*. The Irish component is largely driven by clusters with individuals who have recent genealogical ancestry (i.e., from the 1850s) from the Wexford and Waterford regions of southern Ireland, primarily *N&W Munster* and *SE Leinster*. Both these English and Irish contributions to Irish-British IBD-segment sharing with NL are strikingly supportive of historical records which show the migrants who migrated to NL can be traced back to communities from these regions in Britain and Ireland[3]. Moreover, the contributions from each individual Irish/British cluster to the Irish or British ancestry profile in NL are in similar proportions across NL clusters. This is suggestive of a single source of the Irish and British ancestry in NL, i.e., that the Irish or British ancestry in NL is not from multiple waves from different regions in Ireland or Britain.

The sharing signal in the southwest of NL of southeast Irish haplotypes is further supported in an unsupervised form of the *nnls* method where we consider each NL or Irish-British cluster as a mixture (Supplementary Fig. 29) of shared IBD (separately shown in Supplementary Fig. 30) from any other NL or Irish-British cluster. Results show that whilst most NL clusters predominantly share IBD-contributions from other NL clusters, reflective of their shared ancestry, some clusters such as *Burin E 1* or *Avalon Pen.* still present substantial ancestry contributions from the *N&W Munster* and *SE Leinster* clusters. Furthermore, we evaluated this mixture of Irish and British haplotypes with the *fastGLOBETROTTER* algorithm, whose mixture model agrees with an Irish ancestry source best represented by *N&W Munster* and *SE Leinster* (see Supplementary Note 6).

**Evidence of population bottleneck and homogeneity.** Existing literature from historical[3], clinical[1,10–14], and population genetic studies[4] suggests evidence of a population bottleneck in the European settlement of NL. Furthermore, recent work has shown that island communities tend to experience stronger founder effects[40]. Therefore, we set out to characterise the magnitude of the NL founder effect using the $NL_{1,807}$ dataset and comparing this with European source populations.

We first sought to estimate the historical effective population size ($N_e$) of NL to provide insight to past population bottlenecks. Using our IBD segment sharing data, we applied *IBDNe*[27] to estimate historical $N_e$ in NL clusters >100 individuals in membership, comparing to Irish and British regions (see Methods and Supplemental Note 7 for evaluation of segment accuracy). We observe (Fig. 4a, b, Supplementary Data 4–5) that both Ireland and Britain, and NL have experienced a period of exponential growth within the last 10 generations, consistent with previous estimates of other European ancestry populations[41,42]. Within Ireland and Britain, in general England has a higher $N_e$ than Wales, Scotland, or Ireland, and Orkney. British estimates are consistent with previous estimations from the PoBI dataset[42]. Within NL we detect a consistent reduction of ancestral population size 15-10 generations ago across tested clusters. This population size reduction is several orders of magnitude lower to Irish or British equivalents at a comparable time-period. $N_e$ estimates prior to this reduction are much larger than Irish or British estimates and have wide intervals, perhaps due to the impact of the settlement-bottleneck masking previous demographic profiles. NL regions associated with inter-regional admixture such as *Avalon N* or *Admix NL* have higher $N_e$ estimates within the past 10 generations which would support evidence of an intra-NL admixture history.

To complement this non-parametric modelling, we also recorded and compared sharing of IBD-segments (Fig. 4c, Supp Data 5) and Runs-of-Homozygosity (ROH) (Fig. 4d, Supplementary Data 6) within individual NL clusters and Irish or British clusters. We compared the relationship between average number of, and total length of, IBD-segments >3 cM and <15 cM shared between individuals placed in the same cluster (Fig. 4c) - this would reflect recent background community relationships and degree of isolation[39]. NL clusters consistently present higher levels of IBD sharing than British or Irish clusters, and in some cases (e.g., Trinity and Conception Bay clusters) higher than Orkney and Wales. When compared to Orkney or Wales, NL clusters share slightly fewer segments on average even though the total length shared is comparable. This suggests that the increase of relatedness in NL is more recent than Orkney or Wales, or that on average the $N_e$ is overall higher in NL. Elevated IBD levels within NL are also supported by IBD sharing patterns between NL clusters compared to between Irish or British clusters

(Supplementary Fig. 30–32), showing a general pattern of elevated haplotype sharing across the province as well as within specific genetic communities. Also supportive of a general elevation of haplotype sharing, ROH levels in NL are higher than the average in Ireland or Britain (Fig. 4d), with some NL clusters in Trinity Bay (for example) exhibiting particularly high levels. We also show the equivalent plots per-individual in Supplementary Figs. 62–64. This increase in ROH seems driven by longer ROH, consistent with relatively recent isolation. Some individuals show ROH levels consistent with recent consanguinity, i.e., more than 50 cM of genome covered by ROH > 20 cM in length. We show proportions of these individuals in each NL European-ancestry cluster in Supplementary Fig. 65, further demonstrating recent isolation.

Finally, we further investigated the evidence of NL-specific genetic drift, to inform on the suitability of NL as an ideal study population for enrichment of rare functional variation. Utilising Patterson's $D$ statistic[43], we first confirmed Irish-English comparative affinities in each NL *fineSTRUCTURE* cluster testing $D$ (YRI, NL; Ireland, England) (Fig. 5a, Supplementary Data 7), and where an excess of Irish alleles would result in a negative test statistic. Whilst all clusters show a positive statistic, we find that the four NL clusters identified with Irish ancestry in our haplotype analysis (Fig. 3) are confirmed to present an excess of Irish alleles when compared to English references (Fig. 5a). Next, we tested for NL-specific drift by generating two $D$ test statistics for each NL cluster which together would differentiate shared drift between NL clusters and; (i) Ireland/England, and (ii) other NL individuals (see "Methods"). An excess of NL allele sharing in both tests may indicate NL-specific drift independent of Ireland or England. We find that most NL clusters, but especially *Conception Bay S 2* and *Conception Bay W*, present excess NL drift (Fig. 5b, Supp Data 8). In the case of *Conception Bay S 2* and *Conception Bay W* this is associated with higher IBD sharing within those clusters, which would be consistent with an isolated community experiencing excess genetic drift.

## Discussion

We have performed a systematic analysis of the European ancestry within the Canadian province of NL. Irish and British settlers whose ancestry is shared predominantly with the southeast of Ireland and the southwest of England, respectively, underwent a genetic bottleneck approximately 10–15 generations ago (300–450 years ago assuming generational time of 30 years[27,44,45]). This mixture of south-western English and south-eastern Irish ancestry was distributed unevenly across NL, with Irish ancestry predominant in the south and south-east, and English ancestry predominant elsewhere. Post-migration, genetic structure shows geographic stratification around the bays of the island due to population isolation and then expansion. Genetic homogeneity was elevated with increased haplotype sharing both between and within individuals, resulting in NL-specific drift of allele frequencies. The St John's region of NL in northern Avalon Peninsula is associated with wide-spread haplotype sharing with other regions of NL, suggesting that the urban area was a target of post-settlement migration and resultant admixture from across the province.

Our analysis has substantially expanded our understanding of the genetics of a population with evidence of genetic isolation and bottlenecks. Previous work on rare Mendelian conditions[10–14] in NL has highlighted the potential of the population for genetic mapping efforts, a case explicitly argued for previously[1]. Previous analysis of 494 NL individuals[4] showed initial evidence of isolation and genetic structure based along British/Irish ancestry associated with Christian Catholic/Protestant denomination. We

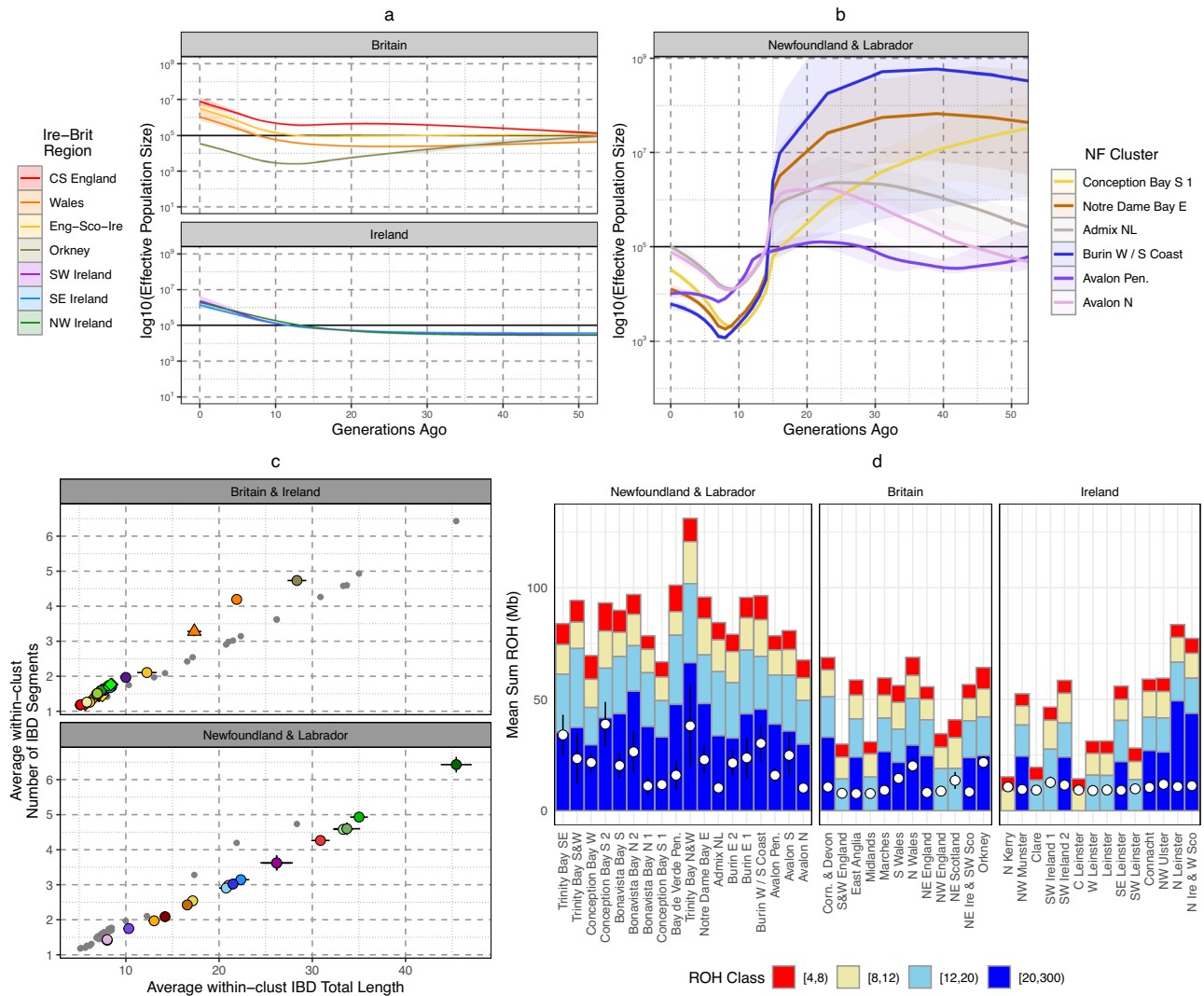

**Fig. 4 Evidence of bottleneck with Newfoundland and Labrador. a** The estimated historical $N_e$ for Irish and British regions using IBD-segments and the *IBDNe* tool. Shading shows the 95% confidence intervals. **b** The estimated historical $N_e$ for NL clusters >100 individuals in membership size. Shading indicates 95% confidence intervals. **c** The individual mean total length of Identity-by-Descent (IBD) segments shared with another individual placed in the same cluster, versus the mean number of IBD segments shared with individuals placed in the same cluster. Clusters are colour and shaped coded in the same system as Fig. 1. Irish and British clusters are plotted separately from NL clusters for legibility, and grey points indicate non-Irish and British (or non-NL) clusters. Error bars show the 95% confidence intervals. **d** The average total length of ROH within each NL or Irish and British cluster. The mean total length of ROH > 1.5 Mb for each cluster is shown by a hollow white circle (left y axis), or the proportion of the genome in ROH > 1.5 Mb in length ($F_{ROH}$—right y axis). Also shown are average total lengths of ROH in four length bins: (1) 4 cM ≤ROH < 8 cM, (2) 8 cM ≤ROH < 12 cM, (3) 12 cM ≤ROH < 20 cM, (4) 20 cM ≤ROH < 300 cM. Groups of clusters are coloured together, and error bars shows 95% confidence intervals. All panels were plotted using the statistical computing language R[63] and the package ggplot2.

show that the NLGP dataset is an equivalent sample of that population structure and diversity - but using a much larger NL sample, more genetic markers and updated statistical approaches, we have extended this work to show, in much finer detail and with higher resolution, the genetic structure across the island of Newfoundland. Identifying this fine-scale structure in NL informs on the potential patterns of rare genetic variation stratification in NL[20,21]. Further, our results on the extent of haplotype sharing and genetic homogeneity in NL show that they are comparable to the Orkney Islands, a well-established isolate[46–48]. Evidence of a bottleneck as well as mixing of English and Irish ancestry is remarkably consistent with the dates of major English/Irish settlement within the 18th and 19th centuries (i.e., consistent with estimates of 10 generations ago). In addition, these results have furthered our understanding of European settler genetic history

in NL which is of interest to genetic genealogy as well as historical research. Our data indicates that a Protestant or Catholic religious background still acts as a proxy for English or Irish ancestry even after 300 years of settlement. The fact that Irish was predominantly spoken in some of these communities until the early-20th century in Newfoundland is consistent with these observations. Whilst confounding factors may influence the co-stratification of Christian religion and population structure, we show that same-denomination cluster pairs do share more haplotypes in common than differing-denomination at the same distances - even in fine-scale on the Burin Peninsula where Catholic and Protestant clusters are geographically interspersed. It is likely that that genetic structure in NL is still consistent with a predominantly isolation-by-distance model—but where religious structure has also played an important role. Lastly, using

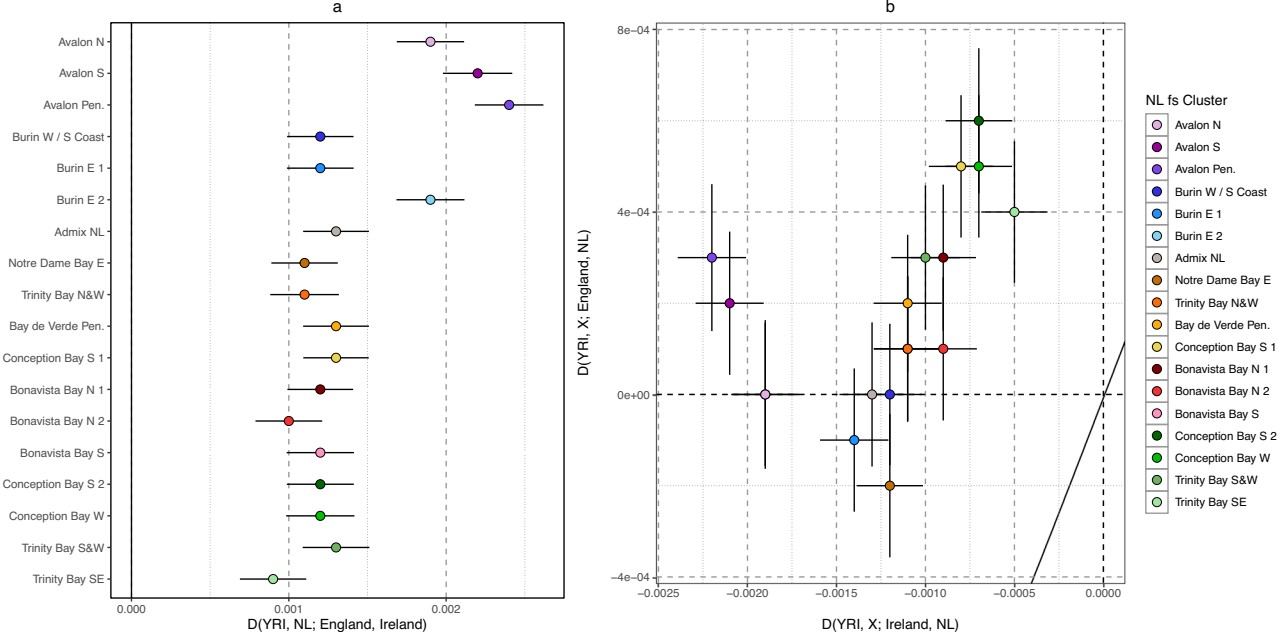

**Fig. 5 Evidence of NL specific genetic drift from Irish and English source. a** Patterson's $D$ statistic[43] testing the allele sharing between NL clusters and English and Irish population references. Positive values indicate excess sharing within NL clusters to Ireland, and negative indicate excess sharing to England. YRI indicates the outgroup for these tests (Yoruban sampled in the 1000 Genomes Project Phase 3). **b** Testing NL-specific drift with Patterson's $D$ statistic. X-axis tests if each NL cluster presents an excess of Irish (negative $D$ value) or NL (positive $D$ value) alleles, and Y axis tests if each NL cluster presents an excess of English (negative $D$ value) or NL (positive $D$ value) alleles. All panels were plotted using the statistical computing language R[63] and the package ggplot2, with error bars showing 1 standard error.

detailed Irish and British ancestry references, we were able to trace these sources to the specific regions of southeast Ireland and southwest England, improving upon the observations of Zhai et al.[4] This work also highlights the use of detailed reference datasets[23,30] with geographic or regional data in elucidating human history.

Surprisingly, within our analysis of Britain and Ireland to identify reference clusters for ancestry estimation, we found our IBD-network method provided greater resolution than *fineSTRUCTURE* in detecting population structure within the south of England[22]. This allowed us to genetically differentiate the communities in the centre and south of England which were grouped into a one equivalent cluster in a previous *fineSTRUCTURE* analysis[22]. This highlights the potential of relatively simple IBD-based clustering methods that not only scale to large datasets but provide insights where other more computationally intensive methods cannot. It is possible that the IBD-based and *Chromo-Painter*-based approaches are accessing subtly different time-periods of coalescing genealogies, hence their differing power to detect structure in different regions of Britain. To further highlight this, although IBD-based methods were able to show structure in the south of England, *fineSTRUCTURE* methods were able to differentiate more clusters in north England than our IBD-based approach[22].

There were some limitations in this work. Without substantial French and specifically French-Canadian references, we were unable to adequately explore the degree of French ancestry within NL, where the southern and western coasts of NL have been associated with French migration. In a limited $D$ statistic analysis (Supplementary Fig. 33), we detected little evidence of excess French allele sharing in any one NL cluster compared to Ireland or Britain. These observations should be considered as preliminary as this utilised a limited sample size and lacked haplotype data which is more sensitive to sub-continental ancestry differences. Future analysis using French Canadian and French

samples may further our understanding of such ancestry to the NL population structure. Moreover, whilst we were able to group individuals by their religious background, we did not have information on self-reported Irish/British background, limiting our ability to detect any communities of Protestant Irish or Catholic English. Finally, the original aim of this study was to investigate the genetic history of the European settlers of NL, particularly from Britain and Ireland. While we detected individuals with varying degrees of Indigenous ancestry and included them in analysis of NL fine-scale structure, given the lack of appropriate North American Indigenous references and the small number of participants (we sampled 73 individuals with putatively admixed Indigenous American ancestry), we made no further effort to characterise indigenous ancestry in NL. Should a study of the genetic history of the Indigenous Peoples within NL be undertaken in the future, it will require a dedicated study design co-developed with representatives from Indigenous communities in order to address the priorities of those communities as recommended[49].

In conclusion, we have highlighted the role of south-western English and south-eastern Irish settlers in forming a genetic landscape with footprints of a substantial bottleneck forming about 10–15 generations ago, consistent with historical records of substantial migration from Ireland and Britain to NL. This bottleneck, and subsequent isolation and expansion of populations around the bays of NL, has shaped a genetic landscape with both high differentiation and haplotype sharing within communities. Results detecting genetic drift shows evidence of NL-specific drift, future work with whole-exome or whole-genome sequence data would help to quantify any drift of rare and functional alleles similarly to that achieved in the isolated island population of Shetland[15]. NL's demographic history has left a unique genetic background and this genetic profile is ideal for a large population-based genetic mapping efforts, like those achieved in Iceland[50], Finland[16], and the Orcadian[48] populations. Our results not only

highlight the use of annotated population genetic references in elucidating human history but also characterise the genetic profile population ideal for human genetic disease mapping in unprecedented detail.

## Methods

**Datasets**. The Newfoundland and Labrador Genome Project (NLGP) is a general population cohort focused on the characterization of Newfoundland and Labrador's population (Sequence Bio Inc., 2021). The NLGP cohort consists of the initial 2,500 participants for whom medical history and saliva samples were obtained with informed consent under a study protocol approved by the Newfoundland and Labrador Health Research Ethics Board (Ethics Reference #: 2018.243). The NL Genome Project was based on the random recruitment of participants from General Practice clinics from across the province that were 18 years of age and over, possessed a valid NL health card, and provided written consent. This may bias towards individuals or communities with existing health-care access. As part of the participant's self-reported data, information was collected on their religion and the birthplace of their parental ancestors. Each participant provided a saliva sample using the DNA Genotek Oragene OG-600 collection kit (DNA Genotek, Ottawa, Canada). DNA extracted from these samples was genotyped using the Illumina Global Diversity Array (GDA; Illumina, San Diego, CA). A quality threshold of 99.2% SNP pass rate per sample resulted in 2,446 individuals and 1,721,246 SNPs passing initial genotyping QC. Marker genome positions were made available in both human genome builds 37 and 38.

Previous NL reference genotypes[4] were accessed from the Gene Expression Omnibus (GEO) accession GSE74392, covering 442 individuals and 1.3 M SNPs and INDELS over the Affymetrix Axiom Genome-Wide Array platform.

Irish and British reference genotypes were assembled from the previously reported Irish DNA Atlas[23], Trinity Student[31], and the People of the British Isles (PoBI)[30] datasets. These three datasets were first merged in a combined dataset of 4,469 individuals and 419,033 common SNPs aligned to human genome build 37.

Genotypes from the combined worldwide references from the 1000 Genomes Project Phase 3 (KGP3)[51] and the Human Genome Diversity Project (HGDP) were accessed from the gnomad v3.1 data release, filtering for autosomal bi-allelic SNPs. A total of 3,942 individuals were used to characterise global patterns of diversity.

**NL-ancestry Identification**. To characterise global ancestry within the NLGP dataset, we combined the genotypes of 2,446 NL individuals with 3,942 individuals from the world-wide reference dataset. We selected individuals and SNPs with missingness proportions <5%, selecting non-A/T or G/C SNPs with a MAF > 2%. To remove markers poorly genotyped, we applied the --hwe 1e-9 midp keep-fewhet filter, removing SNPs which violated the Hardy-Weinberg equilibrium at a p-value significance <1e−9, with a mid-p adjustment recommended[52], as well as keeping markers which failed due to too few heterozygotes as this is expected from significant population stratification (see PLINK 2.0 documentation). After pruning SNPs in strong LD with the parameters --indep-pairwise 1000 50 0.2, we were left with a dataset of 228,761 common SNPs. All filtering was performed with using plink2.0[53,54]. We projected the NL individuals onto the principal components calculated from the world-wide references using the --pca --pca-cluster-names option and removed ancestry outliers based on their separation with non-European or non-American ancestry clusters, leaving a dataset of 2,417 individuals with either European, Indigenous, or mixed European-Indigenous ancestry.

With the ancestry dataset generated above, we re-extracted SNP genotypes, selecting non-A/T or G/C SNPs with missingness <5%, MAF > 2%, and removing SNPs that failed HWE at significance of <1e−9. With this filtered set of individuals and genotypes we estimated familial relatedness with KING and the --related option of relatedness estimation – randomly removing one from each pair of individuals with a 3rd degree relationship or closer. We chose a higher threshold than commonly used for population-based analyses as the subsequent haplotype-based analyses have greater sensitivity to relationship between individuals than so-called "unlinked" analyses[22]. The final dataset consisted of 1,807 individuals and 685,221 common SNPs (the "NL$_{1,807}$" dataset).

**NL population structure**. To perform ChromoPainter[32] based haplotype painting, we first phased the genotypes into inferred haplotypes using SHAPEIT v4[55] with default parameters, using a recombination map from human genome build 38. Using bcftools[56] and scripts provided by the authors of ChromoPainter, we then converted the phased haplotypes from vcf format to the input format required by ChromoPainter.

With haplotype data encoded in ChromoPainter phase and recombrates format, we utilised the fs utility which combines ChromoPainter and fineSTRUCTURE[32] analyses in a single unified pipeline. We estimated the parameters Ne and mu for the calculation of the ChromoPainter co-ancestry matrix which records the number of haplotypes that an individual copies from other individuals. Using a random subset of chromosomes (3, 9, 16, and 21) we estimated Ne and mu in all NL$_{1807}$ individuals. With this estimate (otherwise known as stage 1 in fs), we calculated the co-ancestry matrix, painting all NL$_{1807}$ individuals as a mixture of haplotypes donated from every other NL$_{1807}$ individual (stage 2). For stage 1 and 2, we set the expected haplotype chunks to define a region to 50 rather than the default 100 to

reflect the expected homogeneity in the NL cohort[22]. With the resultant chunkcounts co-ancestry matrix we performed fineSTRUCTURE clustering (stage 3), using 2 million burnin and 2 million sampling iterations of fineSTRUCTURE's MCMC clustering algorithm, sampling states every 4000 iterations. With the MCMC, having sampled the highest posterior probability we performed 200,000 additional tree-building steps (stage 4) to reach a final inferred maximum a posteriori (MAP) state and dendrogram of clusters.

We performed several additional analyses to further characterise the clustering of the NL$_{1807}$ dataset generated by fineSTRUCTURE. We processed and visualised the dendrogram in R in part using scripts provided by the authors of fs.

We mapped the geographic distribution of clusters by plotting the birthplaces of grandparents born within the same region, so that by restricting the analyses to these grandparents mitigated effects from within-NL migration from the 20th century onward. We first calculated the distance between grandparents of the same NL participant, measured by latitude and longitude. For each grandparent pair (between grandparents i and j), we calculated the Euclidean distance (Eq. (1)). Then, calculating the mean distance between all grandparents of the same participant, we filtered for grandparent birthplaces of individuals with a mean grandparental distance ≤0.5.

$$\sqrt{\left(Lat_i - Lat_j\right)^2 + \left(Long_i - Long_j\right)^2} \qquad (1)$$

We visualised the multi-dimensional sharing of the ChromoPainter co-ancestry matrix by performing principal component analysis using R functions supplied by the authors of fs. To visualise the Indigenous ancestry component, and the structure of European ancestry across the province we visualised PC1 versus PC2, and PC2 versus PC3. We estimated genetic distance between clusters using the Hudson F$_{ST}$ estimator[57] implemented in the admixtools2 R package (manuscript in preparation), using the Hudson F$_{ST}$ estimator on a set of 182,222 LD-independent SNPs identified through PLINK using the command --indep-pairwise 1000 50 0.2. We applied the chi-squared test to assess religious association with fineSTRUCTURE clusters, computing p-values with simulation because of low counts in the contingency table using the chisq.test function within R. To further enhance the visualisation of the co-ancestry matrix, we applied the t-SNE and the umap[58] algorithms implemented in the R packages Rtsne and uwot, respectively. With the exception of setting the t-SNE perplexity to 10, we used default parameters to generate the lower dimensional space.

**NL Irish-British ancestry**. To explore the genetic links between NL, and Ireland and Britain, the NL$_{1807}$ dataset was combined with the 4,469 individuals of Irish and British ancestry (see Datasets). Combining these datasets and selecting SNPs with a genotype missingness <5%, MAF > 2% and removing SNPs which failed HWE at significance of <1e−6 resulted in a combined dataset with 178,603 common SNPs aligned to the human genome build 37. The combined dataset was then phased together using SHAPEIT v4 to generate a set of inferred NL, Irish, and British haplotypes. The results of a PCA of this dataset is shown in Supplemental Data 3.

To estimate British and Irish ancestry in NL we used sharing of Identical-by-Descent (IBD) segments. We detected segments, > 1 cM in length, from phased genotypes using refinedIBD[59]. We then combined adjacent segments putatively fragments of a true, larger, IBD segment broken up by genotyping or phasing errors with the merge-ibd-segments utility using default parameters.

To explore regional Irish and British ancestry in an expanded sample of Irish and British genotypes (using more Irish genotypes from the Trinity Student dataset than previously analysed[23]) we decided to cluster Irish and British individuals based on genetic sharing. We leveraged the Louvain community detection algorithm[60] to cluster a network where each individual is a node, and each edge is the total length of IBD in cM that one individual shares with another. We excluded extreme outlier edges which shared >142 cM which are assumed to be more reflective of familial relationships. This approach has been successful in extremely large datasets[37,38], and was used as it scales to our sample size where other methods such as fineSTRUCTURE become a computational burden. The Louvain clustering was performed hierarchically, with three rounds of clustering, each performed on each cluster detected in a previous hierarchical level of clustering. The first level generally split Ireland from Britain, the second level identified broad regions (e.g., NW Ireland, SW Ireland, SE Ireland), and the third level identified the individual clusters reported in Fig. 3a. The membership of each third level cluster to each second level group are as follows: CS England (S&W England, East Anglia, Midlands, Corn.& Devon), N Britain (NE England, NW England, NE Scotland, NE Ire & SW Sco), Wales (Marches, S Wales, N Wales), Orkney (Orkney), SW Ireland (N Kerry, NW Munster, Clare, SW Ireland 1, SW Ireland 2), SE Ireland (C Leinster, W Leinster, Leinster, SE Leinster, SW Leinster), NW Ireland (Connacht, NW Ulster, N Leinster, N Ire& N Wes& SW Sco).

To estimate ancestry, we performed a modification of the nnls methods reported by Leslie et al.[22], here we leverage IBD-segment sharing as a proxy of recent ancestry. We considered IBD segments shared between NL, Ireland, and Britain that were <15 cM and >3 cM, corresponding to approximately 5 to 15 generations (though this is rough approximation[61]). Our modification of the method generated two matrices of X, and Y. X is a matrix of n rows and m columns where each row is the total amount of IBD > 3 cm and <15 cM shared between the

row individual $n_i$ and each column individual $m_j$. Each row is divided by the sum of IBD summarised in that row so that each $X_{ij}$ entry is the proportion of IBD recorded in each row. The second matrix is of $m$ rows and $m$ columns, recording the same for each $Y_{ij}$ individual pair. $X$ summarises the IBD sharing between $n$ target individuals and $m$ source individuals, and $Y$ summarises the IBD-sharing between the $m$ source individuals and themselves. We use $nnls$ to estimate the ancestry proportions explained by these IBD-sharing patterns, modelling each target individual as a mixture of the source individuals.

We performed this modified $nnls$ method to estimate sharing proportions in each NL individual as a target and leveraged every Irish or British individual as a source of IBD sharing. This design modelled NL as a mixture of Irish and British sources, both estimating overall sharing proportions, as well as investigating if certain genetic regions of Ireland and Britain share excess haplotypes with NL, thus allowing the investigation of a specific source of Irish and British ancestry in NL. We used the $cor.test()$ function provided in R to test the significance and strength of the relationship between the Irish and British ancestry proportions with religious denomination. We tested the correlation between cluster-averaged estimated Irish or English ancestry proportion and the proportion of individuals in that cluster reporting a Catholic or Protestant domination (Anglican or United) background.

To perform supervised $ADMIXTURE$[33] analysis and to calculate Patterson's $D$ statistic[43] between NL, Irish and British references, and world-wide outgroups, we merged the NLGP, Irish and British references, and KGP3 and HGDP reference genotypes from the gnomad v3.1 data release together. We selected non-AT/GC SNPs common to all three datasets filtering for SNPs with a missingness <5% and a MAF > 5%. We removed SNPs which violated Hardy-Weinberg Equilibrium (HWE) most likely due to genotyping error using the PLINK[53,54] command --hwe --hwe 1e-9 midp keep-fewhet. We additionally removed SNPs in high LD using the plink command --indep-pairwise 1000 50 0.2 for both sets of analyses. In the supervised $ADMIXTURE$ analysis, we first applied the $ADMIXTURE$ algorithm to just, unrelated, KGP3 individuals ($n = 2508$), setting $k$ ancestral populations to five to capture continental ancestry components. We then performed supervised $ADMIXTURE$ analysis on the 1807 NL individuals, using the $P$ matrix calculated from KGP3 allele frequencies.

To estimate $D$ statistics between populations, we used the $qpDstat$ program (version 970) from the admixtools[62] suite of tools. We only selected KGP3 Yoruban (YRI) reference individuals from the world-wide set, and re-filtered SNPs using the same parameters to reflect this exclusion on non-European or Yoruban ancestry. We defined the "English" reference as British individuals placed within the $S\&W$ $England$ cluster, and "Irish" reference individuals as either $NW$ $Munster$ or $SE$ $Leinster$ Irish individuals as these British and Irish clusters contributed the primary signals of the English and Irish ancestry in haplotype analysis.

**NL haplotype diversity**. We used the IBD-sharing patterns of NL, Ireland, and Britain to estimate recent effective population size ($N_e$) over time using the $IBDNe$ utility[27]. Using IBD segments >4 cM in length between individuals of the same population label, we estimated $N_e$ in Ireland and Britain in the second level IBD clusters (see above). We chose this level as tested groups would have a large enough sample size (> 100 members) for $N_e$ estimates to be robust.

To visual haplotype sharing, we considered IBD-segments that were shared between individuals who were placed in the same third level IBD cluster (in the case of Irish or British individuals) or $fineSTRUCTURE$ cluster (in the case of NL individuals). For each individual, we averaged the number and total length of IBD that that individual shared with another placed in the same cluster. Then for each cluster we calculated the mean and confidence intervals, and then plotted the relationship between the number of segments shared and the total length in Irish and British clusters, and NL clusters.

Complementing the IBD analysis, we utilised the "homozygous-by-descent" (HBD, or ROH) segments detected by $refinedIBD$, utilising segments > 1 cM in length. We detected total amounts of ROH in NL, Irish, and British individuals, first using all segments >1 cM in length (Supplementary Figs. 28–29). We identified outliers' individuals in several clusters; therefore, we excluded these based on an outlier exhibiting more than six standard deviations more than the average total length of ROH for the cluster that they belonged to. This resulted in 26 individuals excluded from cluster-wide averages of ROH sharing, which were calculated over binned lengths (in cM) of; [4,8], [8–12], [12,20], and finally [20,200]. Further, we calculated the average total length of ROH > 1 cM detected in each NL, Irish, or British cluster.

To estimate NL-specific allele drift of frequencies, we estimated two sets of $Patterson's$ $D$ $statistic$ for each NL cluster, using the same methodology in **NL Irish-British Ancestry**. One $D$ $statistic$ arrangement of populations tested the comparative allele shifting between Irish references and 500 random NL individuals not members of the tested cluster X ($D$(YRI, X; Ireland, NL)), and the other arrangement comparing English references to the same 500 random NL subset ($D$(YRI, X; England, NL)) (Fig. 5b). An excess of NL allele sharing in both tests may indicate NL-specific drift independent of Ireland or England.

**Statistics and reproducibility**. The summary statistics presented in the study were all calculated in the statistical computing language R[63] v4.1.3, using Rstudio. When calculating the average of a group we state whether we use the arithmetic mean or median. We calculate standard deviation using the R function sd(), and the standard error from dividing the standard deviation from the squared root of the number of observations in the group (typically sample size). When we have randomly chosen observations we have used the sample() function from R, sampling without replacement unless otherwise specified.

## Data availability

The HGDP[64] and KGP3[51] genotype samples with population meta-data were downloaded from the gnomad v3.1 data release. Previous NL references[4] were accessed from the Gene Expression Omnibus (GEO) accession GSE74392. The genotype and sample meta-data from the Irish DNA Atlas[23], Irish Trinity Student[31], and People of the British Isles[30] datasets were accessed through data sharing agreements with the relevant host institutions. Please contact G.L.C for Irish DNA Atlas enquiries, A.M.M and L.C.B for Trinity Student, and W.B for PoBI enquiries. The NL Genome Project (NLGP) genotype and sample metadata were accessed through a data access agreement as part of a collaboration with Sequence Bioinformatics. In accordance with FAIR data access principles and the consented use of this data by participants, data access requests for scientific research and development should be made to Sequence Bioinformatics (info@sequencebio.com). Supplementary Data 1 records the underlying data for Fig. 1, Supplementary Data 3 records for Fig. 3b, c, Supplementary Data 4 for Fig. 4a, b, Supplementary Data 5 for Fig. 4c, Supplementary Data 6 for Fig. 4d, and Supplementary Data 7 and 8 for Fig. 5a, b, respectively.

## Code availability

We used the following software for the analysis of our dataset: PLINK v1.9 and v2.0, KING v2.2.8, fs v4.1.1, SHAPEIT v4, R v4.1.3, ADMIXTOOLS2, refinedIBD v17Jan20, mergeibd v17Jan20, ibdne v23Apr20, qpDstat v970, fastGLOBETROTTER, EEMS. Code used for the analyses reported in this study can be found on a Github repository[65].

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

## Acknowledgements

We would like to thank all the participants who consented to participate in the New-foundland and Labrador Genome Project for enabling this research. We wish to thank The Centre for Applied Genomics, The Hospital for Sick Children, Toronto, Canada for the genomic services they provided. This work was supported by the National University of Ireland Post-Doctoral Fellowship in the Sciences and Engineering (to E.G.). This publication has emanated from research supported in part by a research grant from Science Foundation Ireland (SFI) under Grant Number 16/RC/3948 and co-funded under the European Regional Development Fund, by FutureNeuro industry partners, and a SFI Career Development Award under grant number 13/CDA/2223. This study makes use of data generated by the People of the British Isles (PoBI) project. A full list of the investigators who contributed to the generation of this data is available from the relevant PoBI papers.

## Author contributions

M.S.P, E.G, and G.L.C conceived and designed the project. G.L.C and M.S.P jointly supervised this work. E.G. designed and performed the majority of the bioinformatic analyses. H.Z curated the NL geographic and demographic data, and M.Mc and S.D aided in data visualisation as well as NL data management. S.M. and R.R contributed bioinformatic support. R.A.L, R.E.M.S, G.M, and J.C.S provided project support and comments. S.O, M.Me, A.M.M, L.C.B, and W.B contributed reference data. E.G, G.L.C, M.S.P, and A.L.S wrote the manuscript, and all authors reviewed the manuscript.

## Competing interests

E.G., and G.C received financial consumable support from Sequence Bioinformatics, Inc. to support research efforts performed during the development of the manuscript. H.Z., M.E.M, S.D., S.M., R.A.L., G.M., R.R., R.E.M.S, and M.S.P. are full time employees and shareholders of Sequence BioInformatics, Inc. J.C.S., and A.S.L. are paid scientific con-sultants employed by Sequence BioInformatics, Inc. M.M, S.O'R., A.M.M., L.C.B, and W.B. declare no competing interests.

**Additional information**

