## [Transparent Peer Review File · Communications Biology]

The Newfoundland and Labrador mosaic founder population descends from an Irish and British diaspora from 300 years ago.REVIEWER COMMENTS

Reviewer #1 (Remarks to the Author: Overall significance):

This manuscript analyzes the population genetic structure of Newfoundland and Labrador (NL) based on genome-wide data from 1,807 individuals and dense reference datasets from Britain and Ireland. Using published modern haplotype sharing methods that provide fine-scale resolution, the authors identify a strong geographic substructure that also correlates with Protestant or Christian denomination. In line with what is known from history, the authors identify that there are close genetic links to South England and South Ireland and that there is a strong bottleneck ca. 10-15 generations (or ca. 200 - 300) years ago.

Generally, the article is very well written and well organized. I could easily follow most major ideas and the flow of logic is also very clear.

The results are original and at an unprecedented resolution (due to the dense new data and fine-scale reference dataset). It is an interesting case study of a bottlenecked island population (such as Iceland, Sardinia, and Ryuku islands). However none of the genetic data will be publicly available (it is data associated with a biotechnological company with commercial interest) and no new computational methods have been shared, thus the interest to others in the general genetic community will be mostly as an interesting case study. As the data is not being made available, it is not generally possible to use this highly interesting dataset in public future research.

Technically, the article is sound and the work is competent, and the major claims are well supported. I have some concerns about the quality of the IBD calls though (see critical point A1 below).

I am not an expert on this region, but much relevant work seems to be cited. There could be some more connections to other case example studies of island/bottlenecked populations (e.g. Sardinia, <https://doi.org/10.1038/s41588-018-0215-8>) or the Ryuku islands (<https://doi.org/10.1093/molbev/msab005> using almost an identical approach). A previous study on NL using genome-wide SNPs too is cited briefly in the introduction (<https://doi.org/10.1038/ejhg.2015.256>), but the novel results here are not discussed in its context (which is relevant, see point B2 below).

Below please find detailed comments, first summarize critical comments (A), then major points (B), and then very minor suggestions (C) which are not "make or break" but the authors hopefully find useful.

A: Critical Points (which are make-or-break pitfalls in my eyes):

A1) Quality of IBD calls

The calling of IBD and ROH here is a black box, no validation of the IBD calls is done. This is particularly problematic as the merged dataset (NL, British, Irish) has only 178,603 SNPs (L413), which is at the extremely low end for IBD analysis. But even in the standard regime with 600K or more SNPs, the 5cM region is already problematic in terms of false-positive and limited power, and uncertain length (see e.g. <https://doi.org/10.1016/j.ajhg.2020.02.012>).

For clustering analysis that is not a severe problem, however, it is for methods such as IBDNe that assume that the input IBD data are all perfectly true. Ideally one checks power and false positive rate e.g. by copying-in approaches (<https://doi.org/10.1371/journal.pbio.1001555>), but I understand that this is technically a lot of effort. To move practically forward, one way could be to evaluate the results by using e.g. 7cM as a cutoff (longer IBD are much more robust to call) and compare how stable results are, and compare the IBD within a cluster to the ROH length distribution (after filtering the close-kin offspring with multiple long ROH) - the IBD and ROH histogram (with respect to length in cM) should roughly correspond to each other when normalized per pair of haplotype - ROH is just IBD within one individual, but much easier to identify with less ambiguity.

A2) Measuring the drift of rare alleles.

The whole paragraph L246-L259 at the end of the results describing the potential of rare alleles becoming common in NL is extremely ad-hoc. That there are some more alleles that are rare in Irish and British at high frequency in NL than under a null model of completely no geographic stratification is barely surprising.

Qualitatively, such an effect will likely show up in any non-British/Irish population. If you want to demonstrate excess NL-specific drift I could e.g. recommend outgroup f_3 statistic matrices and compare them to a baseline of other populations.

Overall, I would even say that at the face value of Fig. 5, the effective N_e value bottleneck of ca. 10,000 individuals for very few generations might not be strong enough for substantial genetic drift at the population level - the differentiation will still be $F_{st} \ll 0.01$.

B: Major Points:

B1) Native American Ancestry

It is stated that Native American ancestry is not the aim of this study, which is understandable given that this is a sensitive topic. Nevertheless, all Native American individuals are included in the full haplotype analysis and e.g. drive the first PC of the haplotype copying matrix. They are excluded only quite downstream excluded based on their haplotype cluster.

However, Native American ancestry is a gradient (as seen in the PCA), and it is not really clear whether a heuristic haplotype cluster sets the cutoff, and whether individuals with substantial Native American ancestry remain in some other clusters (and might influence down-stream analysis such as the IBDNe curves). Based on Supp. Fig. 1.2 the filter is quite stringent but it is hard to evaluate what happens with individuals with, say, 10% Native American ancestry.

There would be methods such as <https://doi.org/10.1371/journal.pgen.1007385> that use IBD sharing on tracts of specific ancestry to circumvent biases in admixed populations. This is not trivial of course, and given the approach of not following up on any Native American ancestry, I would recommend filtering based on qpAdm or Admixture results (which will be very stable as these are different continental ancestries) very early on in the analysis and on an explicit cutoff (>95% European, say), to really only include individuals who have little other ancestry and avoid any bias in downstream analysis.

B2) Citation 3 analyzing genome-wide SNP data from $n=494$ individuals from NL found a very similar structure (population substructure linked to Catholic and Protestant, increased ROH indicative of a bottle-neck). It is therefore very relevant for the work here. However, there is no link between these results to the new results here in the "discussion" section - this older work is only mentioned in the introduction.

B3) The relationship between religious and genetic structure is explored only quite superficially. While there is no overinterpretation in the text (and the authors always speak of correlations), it should be mentioned that some overlap of both structures could be a simple consequence that both genetic and religious structure clusters geographically. To go beyond that statement, one would need to explore whether religious boundaries are especially "barriers" beyond geographic structure.

B4) Other methods: This is not make-or-break and it is not my job as a reviewer to request all possible analyses. But in case the authors find it worth their time and worth a try: A nice geographic tool to visualize geographic structure is EEMs (<https://doi.org/10.1038/ng.346>). The follow-up method allows even to visualize the geographic structure of IBD sharing probing different time depths (<https://doi.org/10.1371/journal.pgen.1007908>) - so seemingly ideally suited for the data here.

B5) Analysis of consanguinity. The ROH analysis is based on all ROH longer than a 1.5mb threshold. That completely masks out the time depth of the signal, or whether there are some individuals with excess long ROH (due to close kinship matings) that drive the signal. I would also recommend measuring ROH in cM, as you do for IBD. For a decent visualization of ROH length classes and thresholds for close-kin offspring see e.g. (<https://doi.org/10.1038/s41467-021-25289-w>).

B6) SNP filtering: You filtered for SNPs that violate Hardy-Weinberg equilibrium (e.g. L353 on world-wide data). However, the geographic structure of variants also makes them violate Hardy-Weinberg, e.g. SNPs that are at high frequency only in one particular geographic region. This is not problematic for PCA or haplotype analysis but can be e.g. for the allele frequency analysis (e.g. the markers at particularly high frequency only in NL get

filtered out). It might be better to filter variants that fail Hardy-Weinberg only in continental European populations for that particular analysis.

C: Minor Points.

C1) In the first paragraph of the introduction (L35-L49), two important citations for demographic history are from genetics papers (1 and 3), which might not be the best original source for historical events. There is also no citation for the initial 25,000 individuals who migrated to NL (in L38).

C2) In L54, there is "limited genetic data", which sounds vague - a side sentence describing what kind of DNA data would be helpful.

C3) In L103 you mention F_{st} on the order of 0.0001 - 0.005, and call that "substantial genetic differentiation". But that is actually surprisingly little, and a value not untypical of intracontinental human F_{st} (intercontinental human F_{st} is on the order of 0.1). So I would suggest calling that only "comparably high genetic differentiation".

C4) L103 "as well as religious." The meaning is clear but there seems to be a missing word.

C5) In L111 you mention PCA results but you do not give a reference to a figure.

C6) "IBD segment" could be at least explained in an extra sentence, so the reader unfamiliar with this genetics jargon gets an understanding of that key concept.

C7) In L229: "Ne estimates prior to this reduction are much larger than Irish or British estimates and have wide intervals". It is a known behavior of IBDNe that the estimates prior to severe bottlenecks the estimates are vastly unreliable, as the bottleneck completely dominates all other signals. Unfortunately, this is only "folklore" knowledge and I cannot think of a decent citation for it.

C8) In L266: You use a generational time of 30 years. That's quite typical for population genetics studies, but there are some citations out there to justify this value and can be cited (use a previous genetic study as a reference point).

C9) You prominently mention that you did not have French samples as a reference point to test for French migration in the South and Western coast of NL (L299-L303). It's not much but there are more than 20 French high coverage genomes in the fully public SGDP (additionally also 20 French Basque). I understand that this number is likely too small for e.g. IBD based analysis but e.g. could be useful for exploratory PCA projections or maybe even "finestructure" analysis.

C10) A map is extremely helpful, but Figure 1 and Figure 3 have some redundancy and could possibly be combined. Also, a geographic scale (e.g. what is 100km) could be helpful for the reader not familiar with this region to get an overview of its spatial extent.

Reviewer #1 (Remarks to the Author: Impact):

See above.

Reviewer #1 (Remarks to the Author: Strength of the claims):

See above.

Reviewer #1 (Remarks to the Author: Reproducibility):

There are two main issues, neither data nor any analysis code is shared.

1) The underlying DNA data is not shared, which does not allow any verification of e.g. the IBD calls or the robustness of the results to other or new analyses.

The data availability statement ("researchers interested in accessing the NL Genome Project data are encouraged to contact Sequence Bioinformatics") does not seem to be compatible with the reporting standards of Nature Journals:

"A condition of publication in a Nature Portfolio journal is that authors are required to make materials, data, code, and associated protocols promptly available to readers without undue qualifications."
(<https://www.nature.com/nature-portfolio/editorial-policies/reporting-standards>)

It is a decision of the editor (as there are cases where this has been deemed okay), but I personally believe that it is very tricky to have companies with commercial interests publishing manuscripts that are effectively not checkable by other researchers and could be completely made up without anyone ever knowing.

2) No code was shared. It is best practice in the field to share all custom code used to run the analysis and create visualizations in this manuscript - e.g. in form of a GitHub repository. This really helps overall reproducibility and technical evaluation of the analysis.

Reviewer #2 (Remarks to the Author: Overall significance):

In this study, the authors report a large cohort of array genotype data from inhabitants of Newfoundland and Labrador (NL) and apply haplotype-sharing-based methods to resolve their genetic history in fine resolution. The analyses are well planned and conducted to maximize information extraction from the available data, resulting in a much higher resolution than the results of a previous study with a similar type of data (Zhai et al, 2016; ref #3 in this manuscript). I have only a few suggestions/questions that I hope may be helpful to further improve this study.

Reviewer #2 (Remarks to the Author: Strength of the claims):

1. Although the main messages of isolation and fine-scaled structure of the coastal NL populations are well supported, I still would like to see how each of these individual communities are connected. The authors already calculated IBD sharing between different NL clusters, so could they describe the connectivity between clusters in terms of IBD sharing? For example, would the clusters sharing the same religious background show higher connectivity than those not given the distance? Also, the authors may consider adding e.g. EEMS analysis (Petkova et al., 2016), if this is useful to show barriers and conduits of gene flow between clusters.

2. The authors highlighted that the haplotype-based methods, such as fineSTRUCTURE and IBD-sharing, have high resolution to detect the fine-scaled genetic structure. However, they did not show a comparison of their results with those of simpler single-marker-based methods, such as PCA and ADMIXTURE. When it comes to clustering, machine learning methods such as t-SNE are also well known to perform well. The authors already used t-SNE and UMAP in this study, so it would be great to compare results between these various methods for readers' interests.

3. The authors discuss the Irish/Catholic vs. English/Protestant contrast as a main factor to explain the genetic difference between NL clusters. While being well convinced by this claim, I find an interesting mismatch between the two nearby clusters, Burin E1 and Burin E2. They are two clusters from the eastern shore of the Burin peninsula with a substantial geographic overlap. In Figure 2B, Burin E2 has a high Catholic proportion (sharing this with the Avalon clusters) while Burin E1 has a high protestant proportion. But in Figure 4B, Burin E1 shows high IBD sharing with Irish (like the Avalon clusters) while Burin E2 show high IBD sharing with English like the other clusters. What makes this interesting conversion between Burin E1 and E2? Is there any historical explanation for this?

Reviewer #2 (Remarks to the Author: Reproducibility):

Details of the methods are provided in the Methods section, but data are not fully publicly available. Please consider providing data in a fully public manner after anonymizing it.

Response to Reviewers

Reviewer 1

Critical Points

- **Point 0** - I am not an expert on this region, but much relevant work seems to be cited. There could be some more connections to other case example studies of island/bottlenecked populations (e.g. Sardinia, <https://doi.org/10.1038/s41588-018-0215-8>) or the Ryuku islands (<https://doi.org/10.1093/molbev/msab005> using almost an identical approach). A previous study on NL using genome-wide SNPs too is cited briefly in the introduction (<https://doi.org/10.1038/ejhg.2015.256>), but the novel results here are not discussed in its context (which is relevant, see point B2 below).
 - *We have expanded our references in the introduction to include more island-based isolated communities and their role in the context of genetic mapping in isolated populations (L84-86). Further we have expanded description of Zhai et al, both in the introduction (L78-80) and in the discussion - comparing our novel results to the previous literature (L321-327)*
- **A1) Quality of IBD calls** - The calling of IBD and ROH here is a black box, no validation of the IBD calls is done. This is particularly problematic as the merged dataset (NL, British, Irish) has only 178,603 SNPs (L413), which is at the extremely low end for IBD analysis. But even in the standard regime with 600K or more SNPs, the 5cM region is already problematic in terms of false-positive and limited power, and uncertain length (see e.g. <https://doi.org/10.1016/j.ajhg.2020.02.012>). For clustering analysis that is not a severe problem, however, it is for methods such as IBDNe that assume that the input IBD data are all perfectly true. Ideally one checks power and false positive rate e.g. by copying-in approaches (<https://doi.org/10.1371/journal.pbio.1001555>), but I understand that this is technically a lot of effort. To move practically forward, one way could be to evaluate the results by using e.g. 7cM as a cutoff (longer IBD are much more robust to call) and compare how stable results are, and compare the IBD within a cluster to the ROH length distribution (after filtering the close-kin offspring with multiple long ROH) - the IBD and ROH histogram (with respect to length in cM) should roughly correspond to each other when normalized per pair of haplotype - ROH is just IBD within one individual, but much easier to identify with less ambiguity
 - We have performed several additional analyses to evaluate the quality of our IBD callset, with a focus on the results from the IBDNe analysis, but also between- and within-cluster sharing. This data has been added to the manuscript as Supplementary Data 7.
 - We thank the reviewer for their suggestion regarding that the reduced SNP-set is responsible for substantial differences in total IBD-sharing, as this would have substantial impact. Testing this (Supplementary Data 7), we do observe that accuracy (measuring by agreement of different SNP-callset counts) is a function of increasing segment length as expected. We have performed an additional IBDNe analysis considering only segments > 7cM and have found little differences in our N_e estimates.
 - These additional analyses provide objective evidence that our IBD-based findings in the joint NL and Irish-British datasets are true reflections of the haplotype sharing and diversity within NL compared to Ireland and Britain.
- **A2) Measuring the drift of rare alleles** - The whole paragraph L246-L259 at the end of the results describing the potential of rare alleles becoming common in NL is extremely ad-hoc. That there are some more alleles that are rare in Irish and British at high frequency in NL than under a null model of completely no geographic stratification is barely surprising. Qualitatively, such an effect will likely show up in any non-British/Irish population. If you want to demonstrate excess NL-specific drift I could e.g. recommend outgroup f3 statistic matrices and compare them to a baseline of other populations.
 - *We thank the reviewer for this comment and suggestion and have incorporated results from a Patterson's D^1 statistic testing for NL specific drift. This is shown in a new figure (Figure 5).*

Results have been incorporated into a new Figure (Figure 5). A description of these new results replaces the previous analysis (L290-303). In short, we use paired D statistics testing individual NL cluster allele sharing between either other NL individuals or England/Ireland. An excess of NL-specific drift over shared drift to both source populations is expected to result in a positive value in both the x-axis and y-axis.

- *Further, we have included a brief discussion on future directions in the study of drifted rare variation in NL using next-generation-sequence data that was not available to this study (L379-381).*

Major Points

- **B1) Native American Ancestry** - It is stated that Native American ancestry is not the aim of this study, which is understandable given that this is a sensitive topic. Nevertheless, all Native American individuals are included in the full haplotype analysis and e.g. drive the first PC of the haplotype copying matrix. They are excluded only quite downstream excluded based on their haplotype cluster. However, Native American ancestry is a gradient (as seen in the PCA), and it is not really clear whether a heuristic haplotype cluster sets the cutoff, and whether individuals with substantial Native American ancestry remain in some other clusters (and might influence down-stream analysis such as the IBDNe curves). Based on Supp. Fig. 1.2 the filter is quite stringent but it is hard to evaluate what happens with individuals with, say, 10% Native American ancestry.
 - *We thank the reviewer for this important issue. To further characterize the degree of Indigenous ancestry we performed a supervised ADMIXTURE analysis with individuals from the 1000 Genomes Phase 3 whole-genome-sequence dataset. We first performed a k=5 ADMIXTURE analysis, assuming 5 continental ancestry groups: African, European, South Asian, East Asian, and American. With the SNP weights for each estimated ancestral component, we estimated these components in the NL_{1,807} dataset, finding a consistent profile across most clusters apart from Admix Eur./Indig. 1 and to a lesser extent Admix Eur./Indig. 4 - with no substantial indigenous American ancestry detected apart from the aforementioned clusters. Results of this additional analysis support the position that there is no evidence of Indigenous ancestry within any of the 18 other clusters used in this study. The exclusion criteria are stringent, but the ancestry of the remaining "admixed" clusters is hard to define and represents a small fraction of the overall dataset.*
 - *These results are reported in Supplemental Figure 2.11*
- **B2) Citation 3** - analyzing genome-wide SNP data from n=494 individuals from NL found a very similar structure (population substructure linked to Catholic and Protestant, increased ROH indicative of a bottle-neck). It is therefore very relevant for the work here. However, there is no link between these results to the new results here in the "discussion" section - this older work is only mentioned in the introduction.
 - *To address this point, we have expanded discussion of the previous study of 494 individuals from NL in both the introduction (L78-80) and discussion (L321-327).*
- **B3) The relationship between religious and genetic structure is explored only quite superficially.** While there is no overinterpretation in the text (and the authors always speak of correlations), it should be mentioned that some overlap of both structures could be a simple consequence that both genetic and religious structure clusters geographically. To go beyond that statement, one would need to explore whether religious boundaries are especially "barriers" beyond geographic structure.
 - *This is a helpful point, and we have made an addition to the Discussion (L361-366) where we caution overinterpretation of this association between religious background and genetic structure, and that it is consistent with a standard isolation-by-distance model.*
- **B4) Other methods:** This is not make-or-break and it is not my job as a reviewer to request all possible analyses. But in case the authors find it worth their time and worth a try: A nice geographic tool to visualize geographic structure is EEMs (<https://doi.org/10.1038/ng.346>). The follow-up method allows even to visualize the geographic structure of IBD sharing probing different time depths (<https://doi.org/10.1371/journal.pgen.1007908>) - so seemingly ideally suited for the data here.
 - *We thank the reviewer for this suggestion, and we have now performed EEMS analysis using the NL ancestry data as a supplementary analysis – reported as Supplementary Data 3. We find that the landscape of gene flow and barriers are consistent with fine-scale structure organised around the bays of Newfoundland, and this additional analysis has supported our observations of haplotype-based genetic structure in the province with a different approach.*

- **B5) Analysis of consanguinity.** The ROH analysis is based on all ROH longer than a 1.5mb threshold. That completely masks out the time depth of the signal, or whether there are some individuals with excess long ROH (due to close kinship matings) that drive the signal. I would also recommend measuring ROH in cM, as you do for IBD. For a decent visualization of ROH length classes and thresholds for close-kin offspring see e.g. (<https://doi.org/10.1038/s41467-021-25289-w>).
 - *We have re-analysed ROH in our dataset, this time using the ROH or “HBD” detected by the refinedIBD program to better fit in with our detected IBD analysis. We have also split this analysis into length bins as outlined in the reviewer’s citation, showing a time depth signal to the patterns of ROH we see in NL, Ireland, and Britain. This has allowed our analysis to show exciting results showing that the elevated haplotype sharing in NL is driven by longer segments consistent with a recent bottleneck rather than historical pre-settlement isolation. These changes are reflected in the text (L287-288), Methods (L555-L563), and additional supplementary figures (S5.5-6).*
- **B6) SNP filtering:** You filtered for SNPs that violate Hardy-Weinberg equilibrium (e.g. L353 on world-wide data). However, the geographic structure of variants also makes them violate Hardy-Weinberg, e.g. SNPs that are at high frequency only in one particular geographic region. This is not problematic for PCA or haplotype analysis but can be e.g. for the allele frequency analysis (e.g. the markers at particularly high frequency only in NL get filtered out). It might be better to filter variants that fail Hardy-Weinberg only in continental European populations for that particular analysis.
 - *With regard to the analyses of the world-wide data (ADMIXTURE and fstat), we have re-analysed this data set with a modified filter of SNPs that violate HWE. Using the additional flags of PLINK 2.0’s `--hwe` function, we have included a flag ‘keep-fewhet’ which is designed to retain SNPs which violate HWE significance in the direction expected from population structure rather than genotyping errors (too few heterozygous calls). We have included this in the relevant Methods section (L414-419)*

Minor Points

- **C1)** In the first paragraph of the introduction (L35-L49), two important citations for demographic history are from genetics papers (1 and 3), which might not be the best original source for historical events. There is also no citation for the initial 25,000 individuals who migrated to NL (in L38).
 - *We have modified the introductory text to include population figures which have a clear historical citation (L56-57).*
- **C2)** In L54, there is "limited genetic data", which sounds vague - a side sentence describing what kind of DNA data would be helpful.
 - *L72 has been modified to reflect the specific nature of these studies more closely.*
- **C3)** In L103 you mention F_{st} on the order of 0.0001 - 0.005, and call that "substantial genetic differentiation". But that is actually surprisingly little, and a value not untypical of intracontinental human F_{st} (intercontinental human F_{st} is on the order of 0.1). So I would suggest calling that only "comparably high genetic differentiation". As also emphasized by other comments in this section, please carefully proofread the manuscript for clarity and grammatical errors.
 - *We have changed L123 to reflect this feedback.*
- **C4)** L103 "as well as religious." The meaning is clear but there seems to be a missing word.
 - *This has been changed.*
- **C5)** In L111 you mention PCA results but you do not give a reference to a figure.
 - *This has been added.*
- **C6)** "IBD segment" could be at least explained in an extra sentence, so the reader unfamiliar with this genetics jargon gets an understanding of that key concept.
 - *We have included a brief description of IBD segments on L198-201.*
- **C7)** In L229: "Ne estimates prior to this reduction are much larger than Irish or British estimates and have wide intervals". It is a known behavior of IBDNe that the estimates prior to severe bottlenecks the estimates are vastly unreliable, as the bottleneck completely dominates all other signals. Unfortunately, this is only "folklore" knowledge and I cannot think of a decent citation for it.
 - *Whilst a citation for this phenomenon is unavailable, we have changed the text in L268-269 to reflect this point.*
- **C8)** In L266: You use a generational time of 30 years. That's quite typical for population genetics studies, but there are some citations out there to justify this value and can be cited (use a previous genetic study as a reference point).

- *We have included an original citation of an average 30 years to a human generation, as well as citations of applying that to the IBDNE method at L309.*
- **C9)** You prominently mention that you did not have French samples as a reference point to test for French migration in the South and Western coast of NL (L299-L303). It's not much but there are more than 20 French high coverage genomes in the fully public SGDP (additionally also 20 French Basque). I understand that this number is likely too small for e.g. IBD based 10analysis but e.g. could be useful for exploratory PCA projections or maybe even "finestructure" analysis.
 - *We have included HGDP French individuals from the gnomad v3 data release of whole-genome-sequence genotype data from the HGDP and 1000 Genomes Phase 3 for an exploratory analysis (see Figure S5.7). This analysis suggests a lack of substantial French ancestry in our sample of NL European ancestry.*
- **C10)** A map is extremely helpful, but Figure 1 and Figure 3 have some redundancy and could possibly be combined. Also, a geographic scale (e.g. what is 100km) could be helpful for the reader not familiar with this region to get an overview of its spatial extent.
 - *We have merged Figure 1 and 3 into a new Figure, 2.*

Reviewer 2

- **1.** Although the main messages of isolation and fine-scaled structure of the coastal NL populations are well supported, I still would like to see how each of these individual communities are connected. The authors already calculated IBD sharing between different NL clusters, so could they describe the connectivity between clusters in terms of IBD sharing? For example, would the clusters sharing the same religious background show higher connectivity than those not given the distance? Also, the authors may consider adding e.g. EEMS analysis (Petkova et al., 2016), if this is useful to show barriers and conduits of gene flow between clusters.
 - **Editors comment: This point would be required for further consideration at Communications Biology.**
 - *Authors Response: We thank the reviewer for this comment and have now characterized the connectivity between NL clusters in terms of IBD sharing and have reported these results as part of the new Supplementary Figure S2.9. We show, with low sample size of per-cluster-pairs, that there is tentative evidence that individuals with the religious denomination, same-denomination (e.g., Catholic or Protestant), share more IBD than the average different-denomination cluster-pair, but this signal is subtle and consistent with isolation-by-distance explaining most of the structure that we observe. We have made reference to this new analysis in L171-172.*
 - *As outlined in response to point B4 from Reviewer 1, we have incorporated the results of an EEMS analysis in a new Supplementary Data 3.*
- **2.** The authors highlighted that the haplotype-based methods, such as fineSTRUCTURE and IBD-sharing, have high resolution to detect the fine-scaled genetic structure. However, they did not show a comparison of their results with those of simpler single-marker-based methods, such as PCA and ADMIXTURE. When it comes to clustering, machine learning methods such as t-SNE are also well known to perform well. The authors already used t-SNE and UMAP in this study, so it would be great to compare results between these various methods for readers' interests.
 - **Editors comment: While we agree that benchmarking might be useful, given that the manuscript is not focused on developing a genomic method, this point would not be required for further consideration at Communications Biology.**
- **3.** The authors discuss the Irish/Catholic vs. English/Protestant contrast as a main factor to explain the genetic difference between NL clusters. While being well convinced by this claim, I find an interesting mismatch between the two nearby clusters, Burin E1 and Burin E2. They are two clusters from the eastern shore of the Burin peninsula with a substantial geographic overlap. In Figure 2B, Burin E2 has a high Catholic proportion (sharing this with the Avalon clusters) while Burin E1 has a high protestant proportion. But in Figure 4B, Burin E1 shows high IBD sharing with Irish (like the Avalon clusters) while Burin E2 show high IBD sharing with English like the other clusters. What makes this interesting conversion between Burin E1 and E2? Is there any historical explanation for this?
 - **Editors comment: This point would be required for further consideration at Communications Biology**
 - *The results of high Irish ancestry in Burin E 1 rather than Burin E 2 were due to a labelling error in the original plotting of the IBD-nnls results. The IBD-nnls itself was not impacted by this, only the subsequent labelling. We have checked all other labels in the manuscript for accuracy.*

Other Comments

- Further we have updated the data availability statement with regards to code as well as genotype availability of the NLGP dataset.

References

1. Reich, D., Thangaraj, K., Patterson, N., Price, A.L. & Singh, L. Reconstructing Indian population history. *Nature* **461**, 489-94 (2009).

REVIEWERS' COMMENTS:

Reviewer #1 (Remarks to the Author: Overall significance):

The authors did a thorough job with their revisions. Their additional analyses and text edits effectively addressed each of my previous concerns. Overall, I believe that the manuscript is exceptionally well-written and yields intriguing insights into fine-scale population structure and traces of founder bottle-necks in Newfoundland. In particular, the high-quality visualizations and the clearly structured writing are outstanding. Thus, I believe this manuscript will serve as a showcase example for future studies of genetic structure within founder populations.

I only have three comments and suggestions left regarding the updated parts of the manuscript, please find them listed below. I stress that none of these are substantial roadblocks at this point.

I want to congratulate the authors on their outstanding work, it is evident that they put a lot of effort into this piece and they should be very proud of this manuscript.

Harald Ringbauer

A1) The labels in the caption and in the figure of ROH in Fig. 4d do not match - there are different length scales mentioned.

Overall, the length category visualization really adds to the ROH visualization. One last suggestion: Maybe the authors flag out "consanguineous" individuals (with an excess of long ROH) to avoid having exceptionally recently related individuals having an effect? (e.g. individuals with more than 50 cM of their genome in extremely long ROH > 20cM, say). Moreover, a Supp. Fig. breaking up the ROH bars per group into ROH bars per individual could be an interesting visualization in the Supplement - showing generally elevated levels of background relatedness. I note that I would understand if the authors would not want to show frequencies of consanguinity, of course - which such a figure would implicitly do.

A2) In Fig. S2.11 (the admixture plot using 1000KG $k=5$ components): Something is a bit off here, shouldn't the European-ancestry NL look very similar to European populations in 1000G (in particular GBR)? However, the NL individuals have substantially less of the "pink" European component, and consistently so.

Moreover, according to other analyses (e.g. the PCAs), there should be substantial variation in Native American ancestry in Adm. Eur. / Ind. Cluster 4, but in this supervised admixture there is only one individual standing out, and that only a little?

A3) The new figure S2.9 is highly interesting, as it shows that IBD between Protestant and Catholic clusters at the same geographic distance is much lower than within cluster pairs of the same denomination. In the main text, the authors argue that it is only subtle and confounded by a low cluster number. While there could be other confounders (e.g. some regions having more IBD with everyone), this signal seems very evident.

One interesting follow-up would be doing this per individual pair in a particularly fascinating pair of clusters: Burin E1 and Burin E2. The individual geographic distributions overlap along the coast of the Burin peninsula (Figure S2.2), but Burin E2 of the clusters is genetically and denomination-wise grouped closely with the "catholic" clusters (Fig. 1 and Fig. 3). It is fascinating that the genetic grouping of individuals in the same geographic region (Burin) ended up effectively distinguishing the religious denominations. I actually suggest that this powerful result could be highlighted even more in the main manuscript.

So as one has both religious groups in the same geographic area, showing the IBD by IBD profiles (e.g. IBD of all pairs of samples within a geographic distance bin; for C-C, P-P, and C-P) could really add to the question of denomination being a particular barrier to gene flow.

Reviewer #2 (Remarks to the Author: Overall significance):

In this study, the authors report a large cohort of array genotype data from inhabitants of Newfoundland and Labrador (NL) and apply haplotype-sharing-based methods to resolve their genetic history in fine resolution. The analyses are well planned and conducted to maximize information extraction from the available data, resulting in a much higher resolution than the results of a previous study with a similar type of data (Zhai et al, 2016; ref #3 in this manuscript).

Reviewer #2 (Remarks to the Author: Impact):

The study in the present form suits well with the scope of Communications Biology.

Reviewer #2 (Remarks to the Author: Strength of the claims):

The authors satisfactorily responded to all of my previous comments, and I do not have further comments.

Reviewer #2 (Remarks to the Author: Reproducibility):

Data and code availability statement is clearly written and provides all necessary information for reproducing the results presented in this study.

Response to Reviewers Comments - Second

Reviewer #1

We thank the review for his kind words regarding the manuscript, and for his feedback which has helped shaped the manuscript (and indeed the study) to its current quality. We have included comments back from his remaining suggestions below.

A1. The labels in the caption and in the figure of ROH in Fig. 4d do not match - there are different length scales mentioned. Overall, the length category visualization really adds to the ROH visualization. One last suggestion: Maybe the authors flag out "consanguineous" individuals (with an excess of long ROH) to avoid having exceptionally recently related individuals having an effect? (e.g., individuals with more than 50 cM of their genome in extremely long ROH>20cM, say). Moreover, a Supp. Fig. breaking up the ROH bars per group into ROH bars per individual could be an interesting visualization in the Supplement - showing generally elevated levels of background relatedness. I note that I would understand if the authors would not want to show frequencies of consanguinity, of course - which such a figure would implicitly do.

- There was a typographical error in the legend of Figure 4 as a result of recording the previous version of ROH analysis. We have modified the text of Figure 4's legend to match the correct analysis bins. We have also omitted the results for the putative indigenous-ancestry clusters from panel 4d.
- We have performed an additional analysis visualising the per-individual rates of each of the HBD bins analysed (as well as the 1-4 cM bin for completeness). Please see below:
 - In the first visualisation we show all individuals in each of the NL *fineSTRUCTURE* clusters, excluding putative indigenous-ancestry clusters, and including individuals calculated to be HBD outliers (see Methods). The effect of these outliers hinders visualisation, so we show this without outliers in the next point.

- Shown below is the results without outliers, which demonstrate the elevated levels of long HBD segments across NL, though some clusters appear to contain more longer HBD than other consistent with our results in Figure 4.

- Further, by way of comparison, we show the equivalent results from Britain and Ireland (see below), which show a dearth of longer HBD in comparison to NL. Outlying Irish or British communities include Orkney, Wales, Cornwall and Devon, and some Northern Irish or South Scottish clusters.

- We also explicitly calculate proportions of consanguinity (more than 50 cM of the genome covered by HBD > 20 cM) in each of the clusters analysed for completeness. Please find below. This is calculated without removal of HBD outliers, and we do observe an elevated level of consanguinity in NL.

- We have incorporated these additional results into Supplemental Figures 59-62 and make reference to them in text in lines 274-278. We have decided to include these results as quantifying the degree of consanguinity resulting from the population's isolation (i.e., not due to cultural choice) is informative on the genetic background of European ancestry in NL. We have omitted the results of the putative Indigenous ancestry clusters due to reasons set out in the main manuscript.

A2) In Fig. S2.11 (the admixture plot using 1000KG k=5 components): Something is a bit off here, shouldn't the European-ancestry NL look very similar to European populations in 1000G (in particular GBR)? However, the NL individuals have substantially less of the "pink" European component, and consistently so.

Moreover, according to other analyses (e.g. the PCAs), there should be substantial variation in Native American ancestry in Adm. Eur. / Ind. Cluster 4, but in this supervised admixture there is only one individual standing out, and that only a little?

- We have redone the supervised ADMIXTURE analysis from the beginning as we agree that these results are inconsistent with our other observations. We find that these ADMIXTURE results are entirely consistent with our other results, and we suspect that there was an error in the original analysis which was carried forward. We have redone this analysis and incorporated these results in the Supplementary Information and repeat them here for ease.

A3) The new figure S2.9 is highly interesting, as it shows that IBD between Protestant and Catholic clusters at the same geographic distance is much lower than within cluster pairs of the same denomination. In the main text, the authors argue that it is only subtle and confounded by a low cluster number. While there could be other confounders (e.g. some regions having more IBD with everyone), this signal seems very evident.

One interesting follow-up would be doing this per individual pair in a particularly fascinating pair of clusters: Burin E1 and Burin E2. The individual geographic distributions overlap along the coast of the Burin peninsula (Figure S2.2), but Burin E2 of the clusters is genetically and denomination-wise grouped closely with the “catholic” clusters (Fig. 1 and Fig. 3). It is fascinating that the genetic grouping of individuals in the same geographic region (Burin) ended up effectively distinguishing the religious denominations. I actually suggest that this powerful result could be highlighted even more in the main manuscript.

So as one has both religious groups in the same geographic area, showing the IBD by IBD profiles (e.g. IBD of all pairs of samples within a geographic distance bin; for C-C, P-P, and C-P) could really add to the question of denomination being a particular barrier to gene flow.

- We thank the reviewer for this feedback, this particular analysis has been very interesting to further investigate. We have modified the language in our results section (lines 128-134) to reflect the signal that we do observe, as well as its discussion (lines 325-331)
- Further, we have investigated the Burin peninsula specifically, which we have incorporated as a separate Supplemental Figure 12 - which we describe and repeat here for ease.

- This figure shows the individual distances between ancestors in panel a and the Total IBD that their descendants share in the present day, and the overall distribution of these distances in panel b. Panel c is a zoomed in visualisation of the specific birthplaces of the ancestors analysed in the analysis. We find that there appears to be heightened IBD sharing between the same *fineSTRUCTURE* clusters at similar close distances compared to sharing across clusters - consistent with our observation across all of NL - though the differences are more subtle. We have updated the results and discussion of this manuscript to reflect this.

Analysis Update

We have also incorporated a new analysis that is unrelated to reviewer feedback but nevertheless we feel is an important update. This is detailed below. We have incorporated these results in the main manuscript and the supplementary information as Supp Figures 10-12, and have highlighted these changes in yellow.

We directly compared our sample of the NL population with a previous, smaller, sample of 442 references reported by Zhai et al¹. Using the Gene Expression Omnibus (GEO) accession GSE74392, we downloaded SNP-microarray genotype data generated on the Affymetrix Axiom Genome-Wide Array platform and merged with the genotype data from the NL Genome Project (NLGP) based on SNPs with the same variant rs identifier.

This common dataset included all individuals from the 1,807 dataset of NL-ancestry from the NLGP and 442 NL references from Zhai et al (henceforth the “Zhai” dataset), over 168,075 common SNPs. The substantially reduced marker set in this comparison precluded incorporation of the Zhai dataset into the main analyses of the NLGP dataset - though did allow the direct comparison of the NL_{1,807} and Zhai datasets.

After standard quality control filtering of sample and marker missingness, minor-allele-frequency, and marker deviation from Hardy-Weinberg-Equilibrium expectations using the same thresholds in the NL_{1,807} analysis (see Methods) this left 2,249 individuals and 167,968 common markers. We first investigated relatedness across the two datasets and found evidence that either the two datasets have sampled the same individual or the corresponding monozygotic twin 4 times, parent-offspring pairs 7 times, and siblings 3 times (Supp Figure 10).

We next compared the population genetics of each dataset to confirm that the NLGP is an equivalent sample of NL with regards to genetic variation. We performed PCA using the implementation within PLINK^{2,3}, pruning markers for linkage disequilibrium to gain an unlinked marker-set - using the PLINK command `--indep-pairwise 1000 50 0.2`. We further detected ROH segments using the PLINK `--homozyg` implementation with the following command: `--homozyg --homozyg-window-snp 50 --homozyg-snp 50 --homozyg-kb 1500 --homozyg-gap 1000 --homozyg-density 50 --homozyg-window-missing 5 --homozyg-window-het 1`.

We find that our NLGP sample of NL genetic ancestry matches well with Zhai dataset in principal component space (Supp Figure 11). Individuals from both datasets overlap over the three axis of variation over principal components 1 and 2, and in the space defined by principal components 3 and 4 we identify individuals from the Zhai dataset which overlap with individuals with putative Indigenous American ancestry from the NLGP dataset - matching observations made by Zhai et al¹ that they observed individuals with “aboriginal ancestry”. We tested statistical evidence of difference between each dataset along any of the four principal components with Mann-Whitney test in R⁴ with the `wilcox.test()` function, and found no evidence of difference along principal component 1 (p-value = 0.106), 2 (p-value = 0.4026), or 3 (p-value = 0.1444). We find evidence of difference along principal component 4 (p-value = 0.000299), though this may be due to the small sample sizes of individuals with Indigenous ancestry in either dataset. Lastly, we compared the profile of Runs of Homozygosity (ROH) between each of our NL_{1,807} *fineSTRUCTURE* clusters and the Zhai dataset (Supp Figure 12) and found a general distribution of ROH in the Zhai sample similar to the distribution along the NLGP clusters.

These results provide demonstrable evidence that our wider results from analysis of the population genetics of Newfoundland and Labrador agree well with, and substantially expand, previous samples and investigation of this population.

Supp Figure 10 - Relatedness across the NLGP and the Zhai datasets of Newfoundland and Labrador ancestry. Each point is a pair of related individuals, with colour indicating the type of relationship. Dup/MZ is either monozygotic twin or duplicate sample, PO stands for parent -offspring, FS for full-sibling, and 2nd-4th the degree of relationship beyond. The shape of the point indicates if the relationship is within or between the two NL-ancestry cohorts.

Supp Figure 11 - Shared genetic structure across the NL Genome Project (NLGP) and the Zhai dataset. The first four principal components estimated from PLINK, showing one versus two (a) and three versus four (b). Individuals from the NLGP are colour coded according to *fineSTRUCTURE* cluster membership, and Zhai individuals are shown as solid black point layered above. Each axis is labelled with the variance explained by that component estimated from the eigenvalues from PLINK.

Supp Figure 12 - Distribution of Runs of Homozygosity across the NL Genome Project (NLGP) and the Zhai dataset. (a) The sum of ROH > 1.5 Mb in length detected by PLINK in NL-ancestry individuals across the 22 *fineSTRUCTURE* clusters detected in the NL_{1,807} dataset versus the Zhai dataset. (b) The ROH data shown in panel a but with axis limits of 100 Mb to show the main distribution of sum ROH across clusters. Boxplots show the median value, with lower and upper hinges showing the 1st and 3rd quartiles. Whiskers show the largest value no further than 1.5 x the Interquartile Range (IQR) from that range. Data points beyond these whiskers are plotted separately as black points.

Reviewers Response References

1. Zhai, G. *et al.* Genetic structure of the Newfoundland and Labrador population: founder effects modulate variability. *Eur J Hum Genet* 24, 1063-70 (2016).
2. Purcell, S. *et al.* PLINK: a tool set for whole-genome association and population-based linkage analyses. *Am J Hum Genet* 81, 559-75 (2007).
3. Chang, C.C. *et al.* Second-generation PLINK: rising to the challenge of larger and richer datasets. *Gigascience* 4, 7 (2015).
4. Team., R.C. R: A language and environment for statistical computing. *R Foundation for Statistical Computing.* (2017).